# `FedSVD`: Adaptive Orthogonalization for Private Federated Learning with LoRA

Seanie Lee[1][*]   Sangwoo Park[1][*]   Dong Bok Lee[1][*]   Dominik Wagner[2]
Haebin Seong[1]   Tobias Bocklet[2]   Juho Lee[1]   Sung Ju Hwang[1,3]

[1]KAIST   [2]Technische Hochschule Nürnberg Georg Simon Ohm   [3]DeepAuto.ai

{lsnfamily02, swgger, markhi}@kaist.ac.kr

## Abstract

Low-Rank Adaptation (LoRA), which introduces a product of two trainable low-rank matrices into frozen pre-trained weights, is widely used for efficient fine-tuning of language models in federated learning (FL). However, when combined with differentially private stochastic gradient descent (DP-SGD), LoRA faces substantial noise amplification: DP-SGD perturbs per-sample gradients, and the matrix multiplication of the LoRA update ($BA$) intensifies this effect. Freezing one matrix (*e.g.*, $A$) reduces the noise but restricts model expressiveness, often resulting in suboptimal adaptation. To address this, we propose `FedSVD`, a simple yet effective method that introduces a global reparameterization based on singular value decomposition (SVD). In our approach, each client optimizes only the $B$ matrix and transmits it to the server. The server aggregates the $B$ matrices, computes the product $BA$ using the previous $A$, and refactorizes the result via SVD. This yields a new adaptive $A$ composed of the orthonormal right singular vectors of $BA$, and an updated $B$ containing the remaining SVD components. This reparameterization avoids quadratic noise amplification, while allowing $A$ to better capture the principal directions of the aggregate updates. Moreover, the orthonormal structure of $A$ bounds the gradient norms of $B$ and preserves more signal under DP-SGD, as confirmed by our theoretical analysis. As a result, `FedSVD` consistently improves stability and performance across a variety of privacy settings and benchmarks, outperforming relevant baselines under both private and non-private regimes. Our code is publicly available at https://github.com/seanie12/fed-svd.

## 1   Introduction

Language models have demonstrated remarkable performance across various tasks [32, 24, 8]. While these models provide strong general capabilities, adapting them to specific domains or tasks typically requires fine-tuning with domain-specific datasets [4]. In real-world deployments, however, training data is frequently fragmented across various organizations or user devices, and strict privacy regulations often prohibit direct data sharing [10]. Federated Learning [FL; 21] provides a viable solution by allowing clients to fine-tune models locally on their private data, while a central server aggregates model updates without accessing raw training data, enabling privacy-preserving collaborative training.

In FL, individual clients often lack the computational and memory capacity required for full fine-tuning of large models, making such approaches impractical. Parameter-efficient fine-tuning addresses this by freezing most model parameters and updating only a small subset, enabling scalable

---

[*]Equal Contribution.

39th Conference on Neural Information Processing Systems (NeurIPS 2025).

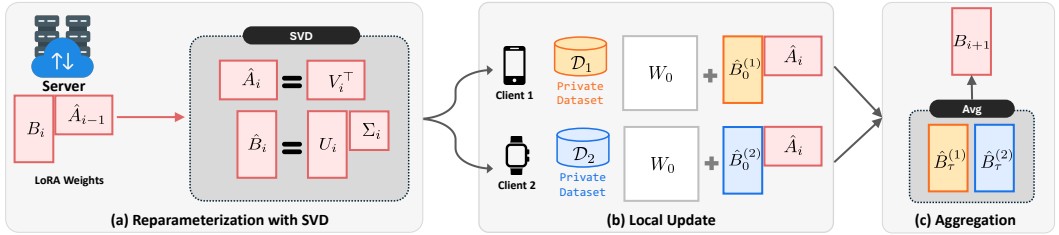

Figure 1: **(a)** At communication round $i$, the server computes the SVD of $B_i\hat{A}_{i-1}$, *i.e.*, $U_i\Sigma_i V_i^\top = B_i\hat{A}_{i-1}$, and initializes $\hat{A}_i = V_i^\top$ and $\hat{B}_i = U_i\Sigma_i$. These reparameterized matrices are then broadcast to all clients. **(b)** Each client updates only the matrix $\hat{B}_0^{(k)}$, initialized with $\hat{B}_i$, using its local dataset, while keeping $\hat{A}_i$ fixed. **(c)** The locally optimized $\hat{B}_\tau^{(k)}$ matrices are aggregated at the server to update the global model.

model adaptation in resource-constrained settings. In particular, Low-Rank Adaptation [LoRA; 16] has been widely adopted for fine-tuning models in FL environments due to its low local computation and communication requirements [40, 31, 12, 34].

Although FL improves privacy by exchanging model updates instead of raw data, it does not provide formal guarantees against information leakage. Sophisticated attacks such as membership inference [27] or model inversion [11], can reconstruct sensitive information from shared updates, particularly given the capacity of language models to memorize training data [6, 7]. Therefore, integrating differential privacy [DP; 9] is essential to provide formal privacy guarantees and enhance the trustworthiness of collaborative model training. A common approach to enforcing DP in deep neural networks is DP-SGD [30, 3, 1], which clips the norm of each per-sample gradient to a predefined threshold and adds Gaussian noise to the average of the clipped gradients.

Recent work [31] has shown that naïve integration of LoRA into DP-SGD significantly degrades model performance. Following a single DP-SGD update of the LoRA adapter matrices $A$ and $B$, the noise added to both matrices is amplified through their product $BA$, as shown in Eq. 5. To mitigate this amplification, FFA-LoRA [31] fixes the $A$ matrix to a randomly initialized constant and updates only the $B$ matrix during training. However, using a fixed random matrix for $A$ limits the learning capability of LoRA, and we observe that optimizing only $B$ leads to significantly slower convergence. Ideally, we would like to adapt $A$ over time to better capture the principal direction of aggregated updates without incurring noise amplification under DP-SGD.

To this end, we propose `FedSVD`, a *simple yet effective* method that introduces global reparameterization based on singular value decomposition (SVD). In the first communication round, the server randomly initializes $A_0$ and $B_0$ and broadcasts them to the participating clients. Each client then optimizes only the matrix $B$ using its local data, and the server aggregates the updated $B$ matrices. In each subsequent round, the server refactorizes the product of the aggregated $B$ and the previous $A$ using SVD to obtain the matrices for the next iteration. As shown in Fig. 1a, the rows of $A$ are re-initialized with orthonormal right singular vectors (*i.e.*, $V_i^\top$) of $BA$ obtained from the SVD. The re-initialization of $B$ uses the remaining components of the SVD, namely the left singular vectors and singular values (*i.e.*, $U_i\Sigma_i$). The newly initialized matrices $\hat{A}_i$ and $\hat{B}_i$ are then broadcast to all clients. Each client $k$, initializes its local matrix $\hat{B}_0^{(k)}$ with $\hat{B}_i$ and subsequently optimizes it to obtain $\hat{B}_\tau^{(k)}$, while keeping $\hat{A}_i$ fixed (Fig. 1b). The resulting $\hat{B}_\tau^{(k)}$ matrices are then collected and aggregated on the server (Fig. 1c).

This SVD-based reparameterization offers several advantages. It allows $A$ to adapt based on the aggregated $B$ without amplifying noise, while maintaining the differential privacy guarantee, since SVD is applied only as a post-processing step after local DP-SGD updates. The orthonormality of $A$ beneficially bounds the gradient norms of $B$, preserving stronger update signals under DP-SGD compared to random initialization. Theoretically, we show that the orthonormal rows of $A$ yield a lower Hessian condition number than a random matrix in a two-layer multilayer perceptron (MLP) with ReLU activations, implying a better-conditioned loss landscape that can potentially lead to faster convergence. Empirically, we observe that this property translates into accelerated accuracy improvement for deep models with orthonormal rows of $A$ (Fig. 3).

We empirically evaluate `FedSVD` on several benchmark datasets, including SNLI [5], MNLI [35], SST2 [29], QQP [26], QNLI [33], and HellaSwag [39], both in private and non-private settings. In

both regimes, `FedSVD` consistently outperforms the relevant baselines during most communication rounds and achieves the highest final accuracy.

We summarize our findings and contributions as follows:

- We propose `FedSVD`, a *simple yet effective* method allowing the LoRA matrix $A$ to adapt over time based on aggregated updates of $B$ using SVD, while eliminating noise amplification under DP-SGD.

- We theoretically show that orthonormal rows of $A$ yield a better-conditioned Hessian of the training loss with respect to $B$ in a two-layer MLP with ReLU.

- We empirically demonstrate that our `FedSVD` approach achieves higher accuracy and faster convergence than relevant baselines under DP-SGD in several benchmark datasets.

## 2  Background

This section reviews the necessary background, including federated learning with LoRA, DP-SGD, and FFA-LoRA. A detailed discussion of related work is deferred to Appendix B.

**Federated learning with LoRA.**  Let $p_\theta : \mathcal{X} \to \mathcal{Y}$ be a language model (*e.g.*, Devlin et al. [8], Liu et al. [20]) parameterized by $\theta$, which maps an input token sequence $\mathbf{x} \in \mathcal{X}$ to an output class label $y \in \mathcal{Y}$. In the FL framework, each client $k \in [K] := \{1, \ldots, K\}$ has access only to its local training dataset $\mathcal{D}_k = \{(\mathbf{x}_i^{(k)}, y_i^{(k)})\}_{i=1}^{n_k}$, where $\mathcal{D}_k \bigcap \mathcal{D}_{k'} = \varnothing$ for all $k, k' \in [K]$ with $k \neq k'$. Furthermore, the central server never accesses any local datasets directly. At each update round $i \in [R]$, a random subset of client indices $S_i \subset [K]$ is selected such that $|S_i| = K'$. Each selected client $k \in S_i$ then receives a copy of the current global model parameters $\theta_i$ from the central server and trains its local model $p_{\theta_i^{(k)}}$ using its private dataset $\mathcal{D}_k$ as follows:

$$\theta_{i,t+1}^{(k)} = \theta_{i,t}^{(k)} - \eta \nabla_\theta \mathcal{L}(\theta_{i,t}^{(k)}; \mathcal{D}_k), \quad \mathcal{L}(\theta_{i,t}^{(k)}; \mathcal{D}_k) = -\frac{1}{n_k} \sum_{(\mathbf{x},y) \in \mathcal{D}_k} \log p_{\theta_{i,t}^{(k)}}(y \mid \mathbf{x}), \quad (1)$$

for $t = 0, \ldots, \tau_k - 1$, where $\eta > 0$ is the learning rate and $\theta_{0,0}^{(k)}$ is initialized with $\theta_i$. Since full fine-tuning of $p_\theta$ is computationally expensive, LoRA is commonly employed to reduce overhead by injecting trainable low-rank matrices into the weight matrix of each layer $l$:

$$W_{i,t}^{(k,l)} = W_0^{(l)} + B_{i,t}^{(k,l)} A_{i,t}^{(k,l)}, \quad (2)$$

where $W_0^{(l)}$ is a frozen pre-trained weight matrix of $p_{\theta_i}$, and $A_{i,t}^{(k,l)} \in \mathbb{R}^{r \times d_{\text{in}}}$ and $B_{i,t}^{(k,l)} \in \mathbb{R}^{d_{\text{out}} \times r}$ are the corresponding low-rank matrices. We denote $\theta_{i,t}^{(k)} = \{(A_{i,t}^{(k,l)}, B_{i,t}^{(k,l)})\}_{l=1}^L$ as the set of LoRA adapter weights for client $k$ at step $t$ of round $i$, where each pair $(A_{i,t}^{(k,l)}, B_{i,t}^{(k,l)})$ represents the LoRA matrices in layer $l$. In methods such as FedAvg [21] and FedIT [40], the server updates its parameters $\theta_i = \{(A_i^{(l)}, B_i^{(l)})\}_{l=1}^L$ by aggregating the weights from the participating clients as follows:

$$A_{i+1}^{(l)} = \left( \sum_{k \in S_i} \frac{n_k}{m_i} A_{i,\tau_k}^{(k,l)} \right), \quad B_{i+1}^{(l)} = \left( \sum_{k \in S_i} \frac{n_k}{m_i} B_{i,\tau_k}^{(k,l)} \right), \quad (3)$$

where $m_i = \sum_{k \in S_i} n_k$ and $n_k = |\mathcal{D}_k|$. At round $i + 1$, the central server model $p_{\theta_{i+1}}$ uses the updated weight matrix for each layer $l \in [L]$ as follows:

$$W_{i+1}^{(l)} = W_0^{(l)} + B_{i+1}^{(l)} A_{i+1}^{(l)}, \quad (4)$$

where $W_0^{(l)}$ denotes the frozen pre-trained weights and $B_{i+1}^{(l)} A_{i+1}^{(l)}$ is the aggregated low-rank update.

**Differential privacy.**  Language models tend to memorize training data, which can lead to the leakage of private information from local client datasets [6, 7]. Differential privacy [**DP**; 9] provides a formal privacy guarantee by limiting the influence of any individual data point on the model, thus mitigating such leakage risks.

**Definition 2.1** (($\epsilon, \delta$)-DP)**.** *A randomized algorithm $M$ is ($\epsilon, \delta$)-differentially private if, for all neighboring datasets $\mathcal{D}, \mathcal{D}'$ that differ in exactly one entry, and all subsets $E$ of the possible outputs of $M$, we have $Pr(M(\mathcal{D}) \in E) \leqslant e^\epsilon Pr(M(\mathcal{D}') \in E) + \delta$, where $\epsilon$ is the privacy budget, and $\delta$ bounds the probability that the privacy loss exceeds $\epsilon$, i.e., the probability that the DP guarantee may fail.*

In FL, privacy guarantee depends on whether the central server is trusted. In the centralized DP setting, clients send raw updates without local privacy measures, and DP is applied during global aggregation [22]. In the local DP setting, which assumes an untrusted server, each client ensures that its update is differentially private before communication [36, 18, 25]. Our work adopts this stronger local DP setting: we apply DP at the client level so that any shared updates (*i.e.*, model parameters) are already privatized. By the post-processing invariance property of DP [9, Proposition 2.1], the final global model also satisfies DP.

**Fixed LoRA $A$ matrix.** A common approach to ensuring the differential privacy of deep neural networks is DP-SGD [30, 3, 1]. DP-SGD first clips each per-sample gradient $g(\mathbf{x}_i)$ from a sampled mini-batch to have a bounded norm by applying $g(\mathbf{x}_i)/\max(1, \|g(\mathbf{x}_i)\|_2/C)$, where $C$ is a predefined threshold. Gaussian noise $\xi \sim \mathcal{N}(0, \sigma^2 C^2 I)$ is then added to the average of the clipped gradients, and the resulting noisy average is used to update the model parameters. However, jointly updating and aggregating both $A$ and $B$, introduces a challenge for fine-tuning models with DP-SGD. During client-side fine-tuning, Gaussian noise is added to the average of the clipped gradients of $A$ and $B$, which becomes amplified through their post-update matrix product after a single DP-SGD step:

$$(B_{i,t}^{(k,l)} + \xi_B^{(k,l)})(A_{i,t}^{(k,l)} + \xi_A^{(k,l)}) = B_{i,t}^{(k,l)} A_{i,t}^{(k,l)} + \xi_B^{(k,l)} A_{i,t}^{(k,l)} + B_i^{(k,l)} \xi_A^{(k,l)} + \xi_B^{(k,l)} \xi_A^{(k,l)}, \quad (5)$$

where $\xi_A^{(k,l)}$ and $\xi_B^{(k,l)}$ represent the Gaussian noise added by DP-SGD. To mitigate the noise amplification caused by the LoRA matrix product, **FFA-LoRA** [31] fixes $A$ as a randomly initialized matrix and performs aggregation only on $B$:

$$W_{i+1}^{(l)} = W_0^{(l)} + \left( \sum_{k \in S_i} \frac{n_k}{m_i} B_{i,\tau_k}^{(k,l)} \right) A_{\text{fixed}}^{(l)}. \quad (6)$$

This removes the quadratic noise term in Eq. 5 (*i.e.*, $\xi_B^{(k,l)} \xi_A^{(k,l)}$), as well as $\xi_A^{(k,l)}$, thus stabilizing model training under DP-SGD. However, using a fixed random matrix $A_{\text{fixed}}^{(l)}$ can affect LoRA learning capacity, potentially leading to suboptimal performance.

## 3  Method

Although FFA-LoRA mitigates noise amplification by freezing $A$, this can lead to suboptimal adaptation, as a fixed random projection may not align well with the data distribution or the dynamics of local model updates. Ideally, $A$ should adapt over time to better capture the principal directions of aggregated updates, while avoiding noise amplification under DP-SGD.

**Periodic re-initialization of $A$ via SVD.** To this end, we propose `FedSVD`, a *simple yet effective* approach that avoids direct optimization of $A$ by periodically resetting it to a new matrix with orthonormal rows, obtained via SVD of the aggregated product $BA$. Specifically, before broadcasting the newly aggregated matrix $B_i$ to the participating clients, the server computes the SVD of $B_i \hat{A}_{i-1}$, where $\hat{A}_{i-1}$ is the matrix from the previous round $i-1$, and initializes $\hat{A}_i$ and $\hat{B}_i$ as follows:

$$\hat{B}_i := U_i[:,:r]\Sigma_i[:r,:r], \quad \hat{A}_i := V^\top[:r,:], \quad U_i \Sigma_i V_i^\top = B_i \hat{A}_{i-1}, \quad (7)$$

where $\hat{B}_0 = \mathbf{0}$, $\hat{A}_0$ is initialized with Kaiming uniform [14], $M[:,:r]$ and $M[:r,:]$ denote the first $r$ columns and rows of the matrix $M$, respectively. Note that we omit the superscript $l$ for brevity. Each client $k$ receives $\hat{A}_i$ and $\hat{B}_i$, and optimizes only $\hat{B}_i$, using Eq. 1 on its local dataset $\mathcal{D}_k$. The server then aggregates the optimized $\hat{B}_i$ matrices from all participating clients. We outline our complete method in Alg. 1.

Importantly, this reparameterization does not change the value of $B_i \hat{A}_{i-1}$, *i.e.*, $B_i \hat{A}_{i-1} = \hat{B}_i \hat{A}_i$, since $\text{rank}(B_i \hat{A}_{i-1}) \leqslant r$ follows from the low-rank structure of LoRA. Therefore, the rank-$r$ SVD exactly recovers $B_i \hat{A}_{i-1}$. As a result, all clients receive a consistent, globally synchronized initialization after SVD, while benefiting from updated, data-informed $\hat{A}$ matrices instead of relying on a fixed random projection. As shown in Sec. 4, this strategy empirically stabilizes training and accelerates optimization.

**Algorithm 1** FedSVD

1: **Input:** Pre-trained language model $p_\theta$, client datasets $\{\mathcal{D}_k\}_{k=1}^K$, total optimization rounds $R$, learning rate $\eta$, batch size $b$, rank $r$, the number of participating clients $K'$.
2: **for** $i = 0, \ldots, R-1$ **do**
3:     **for** for $l = 1, \ldots, L$ **do**             ▷ Broadcast global parameters
4:         **if** $i > 0$ **then**
5:             $U_i, \Sigma_i, V_i^\top \leftarrow \text{SVD}(B_i^{(l)} \hat{A}_{i-1}^{(l)}), \hat{B}_i^{(l)} \leftarrow U_i[:, :r]\Sigma_i[:r, :r], \hat{A}_i^{(l)} \leftarrow V_i^\top[:r, :]$
6:         **else**
7:             $\hat{B}_0^{(l)} \leftarrow \mathbf{0}, \hat{A}_0^{(l)} \leftarrow \texttt{Kaiming\_Uniform}(-d, d)$
8:         **end if**
9:     **end for**
10:     Sample a set of clients $S_i \subset \{1, \ldots, K\}$ with $|S_i| = K', m_i \leftarrow 0$.
11:     **for** each client $k \in S_i$ **do**             ▷ Done in parallel
12:         Initialize the client parameter $\theta_{i,0}^{(k)} = \{(\hat{A}_{i,0}^{(k,l)}, \hat{B}_{i,0}^{(k,l)})\}_{l=1}^L \leftarrow \{(\hat{A}_i^{(l)}, \hat{B}_i^{(l)})\}_{l=1}^L$.
13:         Optimize $\{\hat{B}_{i,0}^{(k,l)}\}_{l=1}^L$ on $\mathcal{D}_k$ with SGD for $\tau_k$ steps with Eq. 1.
14:         $n_k \leftarrow |\mathcal{D}_k|, m_i \leftarrow m_i + n_k,$
15:     **end for**
16:     **for** $l = 1, \ldots, L$ **do**             ▷ Aggregation of parameters updated by the clients
17:         $B_{i+1}^{(l)} \leftarrow \sum_{k \in S_i} \frac{n_k}{m_i} \hat{B}_{i,\tau_k}^{(k,l)}$
18:     **end for**
19: **end for**

---

**Bounding the gradient norm.** Moreover, the orthonormality of $\hat{A}$ ensures that its spectral norm is exactly 1, which leads to a tighter bound on the gradient norm of $B$. Denoting the output as $\mathbf{z} = (W_0 + B\hat{A})\mathbf{x}$, we compute:

$$\left\| \frac{\partial \ell(\mathbf{z})}{\partial B} \right\|_F = \left\| \frac{\partial \ell(\mathbf{z})}{\partial \mathbf{z}}(\hat{A}\mathbf{x})^\top \right\|_F = \left\| \frac{\partial \ell(\mathbf{z})}{\partial \mathbf{z}} \right\|_2 \cdot \|\hat{A}\mathbf{x}\|_2 \leqslant \left\| \frac{\partial \ell(\mathbf{z})}{\partial \mathbf{z}} \right\|_2 \cdot \|\hat{A}\|_2 \cdot \|\mathbf{x}\|_2 = \left\| \frac{\partial \ell(\mathbf{z})}{\partial \mathbf{z}} \right\|_2 \cdot \|\mathbf{x}\|_2, \tag{8}$$

where $\ell(\mathbf{z})$ is a loss function with the corresponding label $y$, $\|\cdot\|_F$ is the Frobenius norm, $\|\hat{A}\|_2$ is the spectral norm of $\hat{A}$, and $\|\mathbf{x}\|_2$ is the $l_2$-norm of $\mathbf{x}$. Under DP-SGD, each per-example gradient is clipped to a fixed norm before noise addition. Thus, any implicit amplification introduced by $\hat{A}$ directly increases the amount of clipping, distorting the original gradient, and weakening the update signal. Since $\|\hat{A}\|_2 = 1$, the gradients reach the clipping threshold with minimal norm, preserving a more genuine update signal under a given privacy budget. In contrast, random initializations usually yield $\|A\|_2 > 1$, necessitating more aggressive clipping and slowing optimization.

**Privacy guarantee of** FedSVD**.** Due to the post-processing invariance property of DP [9, Proposition 2.1], FedSVD guarantees DP by design, as SVD is applied only after $B$ has already been privatized.

**Corollary 3.1** (Privacy guarantee). *By Theorem 1 and the moment accountant from Abadi et al. [1], FedSVD with DP-SGD and FedAvg aggregation satisfies $(\epsilon, \delta)$-DP, given a sampling rate $q$, the total number of local updates $T = \tau R$ per client, and a noise multiplier $\sigma \geqslant c \cdot q\sqrt{T \log(q/\delta)}/\epsilon$ for some constant $c$.*

*Proof.* This is a direct application of the post-processing invariance property of DP [9, Proposition 2.1] and Theorem 1 in Abadi et al. [1].     $\square$

**Theoretical analysis.** We analyze how reparameterizing $A$ and $B$ via an SVD affects the optimization dynamics of $C$-class classification. Consider a labeled dataset $\mathcal{D}_k = \{(\mathbf{x}_i, \mathbf{y}_i)\}_{i=1}^{n_k}$ with one-hot labels $\mathbf{y}_i \in \{0, 1\}^C$. Let

$$W_1 \in \mathbb{R}^{d_h \times d_x}, \quad A \in \mathbb{R}^{r \times d_x}, \quad B \in \mathbb{R}^{d_h \times r}, \quad W_2 \in \mathbb{R}^{C \times d_h}, \tag{9}$$

be parameters of the classification model. With these parameters, let $\mathbf{h}_i = (W_1 + BA)\mathbf{x}_i \in \mathbb{R}^{d_h}, \mathbf{z}_i = W_2 \operatorname{ReLU}(\mathbf{h}_i) \in \mathbb{R}^C$, and $\mathbf{p}_i = \operatorname{softmax}(\mathbf{z}_i)$. We define the cross-entropy loss (with element-wise logarithm) $\mathcal{L}_k(B; A) = \frac{1}{n_k} \sum_{i=1}^{n_k} (-\mathbf{y}_i^\top \log \mathbf{p}_i)$. Let $H_k(B; A)$ be the Hessian of $\mathcal{L}_k(B; A)$ with respect to $B$. Set $\mathcal{A} = A \otimes I_{d_h}$ and, for each $i$, let $S_i = \operatorname{diag}(\mathbf{p}_i) - \mathbf{p}_i \mathbf{p}_i^\top \succeq 0$

and $D_i = \text{diag}\big(\mathbb{1}\{\mathbf{h}_i > 0\}\big)$, where $\mathbb{1}$ denotes elementwise indicator function and $\otimes$ denotes the Kronecker product. Then

$$H_k(B; A) = \mathcal{A}\mathcal{M}_k\mathcal{A}^\top, \qquad \mathcal{M}_k = \frac{1}{n_k}\sum_{i=1}^{n_k}\big(I_{d_h} \otimes \mathbf{x}_i\big)\big(D_iW_2^\top S_iW_2D_i\big)\big(I_{d_h} \otimes \mathbf{x}_i^\top\big). \quad (10)$$

**Proposition 3.2.** *Assume $A$ has full row rank. Then the condition number of the Hessian satisfies*

$$\kappa_2\big(H_k(B; A)\big) \leqslant \kappa_2(A)^2\frac{\lambda_{\max}(\mathcal{M}_k)}{\lambda_{\min}\big(\mathcal{M}_k|_{\mathcal{R}(\mathcal{A}^\top)}\big)}, \quad (11)$$

*where $\lambda_{\min}(\cdot)$ and $\lambda_{\max}(\cdot)$ denote the smallest and largest eigenvalues of a symmetric matrix. If the rows of $A$ are orthonormal (so $\kappa_2(A) = 1$), the bound tightens to*

$$\kappa_2\big(H_k(B; A)\big) \leqslant \frac{\lambda_{\max}(\mathcal{M}_k)}{\lambda_{\min}\big(\mathcal{M}_k|_{\mathcal{R}(\mathcal{A}^\top)}\big)}. \quad (12)$$

The proof is deferred to Appendix A. By reparameterizing using the SVD of $BA$, we write $BA = U\Sigma V^\top$ and choose $\hat{A} = V^\top[: r, :]$ (the top $r$ rows of $V^\top$). Then the rows of $\hat{A}$ are orthonormal, hence $\sigma_{\max}(\hat{A}) = \sigma_{\min}(\hat{A}) = 1$ and $\kappa_2(\hat{A}) = 1$. This removes the $\kappa_2(A)^2$ factor that appears with a fixed random $A$. In contrast, for a randomly initialized $A_{\text{fixed}}$ (*e.g.*, Gaussian, or uniform distribution), its condition number satisfies $\kappa_2(A_{\text{fixed}}) > 1$ with high probability. A smaller Hessian condition number generally indicates a better-conditioned optimization landscape, leading to faster and more stable gradient-based updates to $B$. Thus, our SVD-based reparameterization improves the stability of local client optimization steps by promoting a well-conditioned projection matrix $A$. To directly compute the actual condition number, we use a *simple logistic regression* and show that enforcing the orthonormal structure of $A$ yields a lower condition number (see Table 9 in Appendix D).

## 4 Experiments

In this section, we empirically validate the effectiveness of `FedSVD`.

### 4.1 Experimental Setups

**Datasets.** Following FFA-LoRA [31], we use five datasets, including four from the GLUE benchmark [33]: Stanford Natural Language Inference [**SNLI**; 5], a sentence-pair classification task for textual entailment with three labels (entailment, neutral, contradiction), *i.e.*, NLI task (or recognizing textual entailment); Multi-Genre Natural Language Inference [**MNLI**; 35], the same NLI task, evaluated on both matched (in-domain) and mismatched (cross-domain) test sets; Stanford Sentiment Treebank v2 [**SST-2**; 29], a single-sentence sentiment classification task with two labels (positive, negative); Quora Question Pairs [**QQP**; 26], a paraphrase detection task with two labels (duplicate, not duplicate); and Question Natural Language Inference [**QNLI**; 33], a binary classification task with two labels (entailment, not entailment) that determines whether a context sentence answers a given question. We use the validation split for evaluation, as test splits are unavailable for all datasets except SNLI, which is evaluated on its test split. See Table 7 in Appendix C for the dataset statistics.

**Baselines.** We compare our method, `FedSVD`, against the following baselines:

1. **FedAvg** [21, 40]: Both $A$ and $B$ matrices are fine-tuned locally and averaged independently, as described in Eq. 3.

2. **FFA-LoRA** [31]: The $A$ matrices are initialized with `Kaiming_Uniform`$(-d, d)$ [14] and remain fixed during training. Only the $B$ matrices are fine-tuned and aggregated.

3. **FLoRA** [34]: Both $A$ and $B$ matrices are fine-tuned locally and aggregated by stacking the individual matrices from all clients, rather than averaging them independently. The central server computes the product $BA$ from the stacked matrices and adds it to the pre-trained weight matrix $W_0$. After aggregation, randomly re-initialized $A$, $B$ and updated $W_0$ are sent back to the clients.

4. **FedEx-LoRA** [28]: Both $A$ and $B$ matrices are fine-tuned and aggregated individually as described in Eq. 3. The residual, which is defined as the difference between the aggregated $BA$ and the product of the aggregated $B$ and fixed $A$, is added to the frozen pre-trained matrix $W_0$.

Table 1: Results on 6 GLUE tasks **without privacy constraints**. We report average accuracy and 95% confidence intervals over 5 runs. The best/second-best results are highlighted in **bold**/underline, respectively.

| Method | SNLI | MNLI | | SST-2 | QQP | QNLI | Average |
| | | Matched | Mismatched | | | | |
| --- | --- | --- | --- | --- | --- | --- | --- |
| FedAvg | 84.16 $\pm$ 8.02 | 74.79 $\pm$14.92 | 75.09 $\pm$15.04 | 85.89 $\pm$12.12 | 61.75 $\pm$10.06 | 71.40 $\pm$12.78 | 75.51 $\pm$ 6.61 |
| FFA-LoRA | 82.54 $\pm$ 2.13 | 82.75 $\pm$ 1.72 | 83.45 $\pm$ 1.84 | 94.06 $\pm$ 0.18 | 78.00 $\pm$ 3.08 | 86.61 $\pm$ 1.22 | 84.57 $\pm$ 0.99 |
| FLoRA | 62.17 $\pm$13.26 | 50.49 $\pm$14.93 | 50.81 $\pm$15.27 | 58.99 $\pm$12.47 | 57.91 $\pm$ 7.31 | 62.16 $\pm$10.41 | 57.09 $\pm$ 9.26 |
| FedEX-LoRA | 70.08 $\pm$11.06 | 56.85 $\pm$14.41 | 57.74 $\pm$14.81 | 59.43 $\pm$12.44 | 64.86 $\pm$ 2.39 | 64.90 $\pm$ 12.84 | 62.31 $\pm$ 4.06 |
| FedSVD (ours) | **85.70** $\pm$ 1.23 | **83.96** $\pm$ 2.12 | **84.32** $\pm$ 2.27 | **94.26** $\pm$ 0.51 | **79.82** $\pm$ 2.43 | **88.98** $\pm$ 1.43 | **86.18** $\pm$ 1.44 |

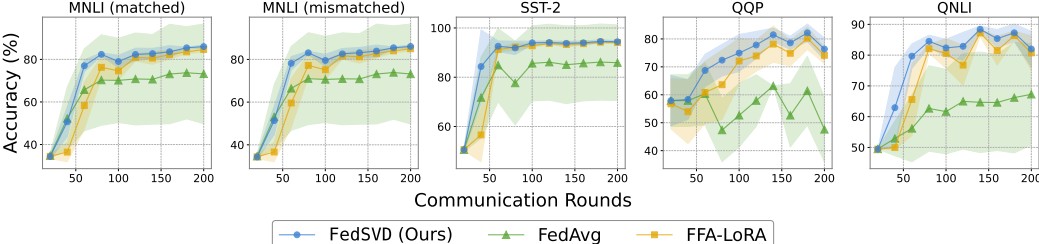

Figure 2: Accuracy vs. communication rounds **without privacy constraints** across 5 GLUE tasks. Curves show average accuracy over 5 runs, with shaded regions indicating 95% confidence intervals.

**Data distribution.** Following Hsu et al. [15], we sample client data proportions from a Dirichlet distribution, with concentration parameter $\alpha = 0.5$ (except in Fig. 4a) for non-i.i.d data. Unless stated otherwise (Fig. 4b), we use six clients in total ($K = 6$). To better emulate realistic federated settings, only half of the clients are randomly sampled for participation in each communication round ($K' = 3$). See Table 8 in Appendix C for per-label distribution across six clients with $\alpha = 0.5$.

**Implementation details.** Following FFA-LoRA [31], we use RoBERTa-large [20] as a base model and apply LoRA [16] with rank $r = 8$ and scaling factor $\alpha = 8$ to the query and value projections, using a LoRA dropout rate of $0.05$. All non-LoRA parameters, including the classification head, are frozen. We run $R = 100$ communication rounds, with participating clients in each round updating their weights using vanilla SGD for $\tau = 10$ local steps. Due to the absence of separate validation splits (except for SNLI), we refrain from extensive hyperparameter tuning. Instead, we adopt values that work reasonably well for FedAvg: learning rate $\eta = 0.5$, clipping norm $C = 2$, and $\delta = 10^{-5}$. The same hyperparameters are applied to all methods for a *fair comparison*. We consider two privacy budgets, $\epsilon \in \{3, 6\}$, where we use the Opacus library [37] to compute the noise multiplier $\sigma$ for a total $T = R \times \tau$ training steps. We use 3 NVIDIA RTX A6000 GPUs for all experiments.

## 4.2 Main Results

**Effectiveness of FedSVD without privacy constraints.** We first assess FedSVD on the GLUE benchmark in a non-private setting. In Table 1, FFA-LoRA outperforms FedAvg, which we attribute to the reduced aggregation error. In contrast, FLoRA, which transmits a large number of parameters, underperforms due to the frequent random re-initialization of the $A$ and $B$ matrices in our experimental setups. We observe a similar pattern in FedEX-LoRA. The proposed FedSVD further improves the performance by periodically adapting $A$ through SVD of the product $BA$ rather than using a fixed $A$. As a result, FedSVD achieves the *highest average accuracy*, outperforming the second-best baseline (FFA-LoRA) by **+1.29 percentage points** (pp). Fig. 2 illustrates accuracy as a function of communication rounds for FedAvg, FFA-LoRA, and FedSVD. FedSVD consistently outperforms the baselines across all rounds. This robustness to early stopping makes FedSVD well-suited for scenarios with limited communication budgets or uncertain convergence points.

**Effectiveness of FedSVD with DP-SGD.** We next evaluate the performance of FedSVD under DP constraints ($\epsilon \in \{3, 6\}, \delta = 10^{-5}$). Table 2 shows that the average gain of FedSVD over FFA-LoRA increases substantially in the DP settings, *i.e.*, from +1.29 pp without privacy constraints to **+8.77 pp** with $\epsilon = 6$. Even under a stricter privacy budget ($\epsilon = 3$), where the injected noise intensifies and the signal-to-noise ratio of gradients degrades notably, our method still achieves an accuracy improvement of **+9.63 pp**, demonstrating its robustness to tighter DP constraints. We attribute this improvement to the SVD-based re-initialization of FedSVD which allows $A$ to capture the principal directions of the aggregated updates more reliably. Furthermore, orthonormal rows of $A$ bound the

Table 2: Results on 6 GLUE tasks **with DP** ($\epsilon \in \{3,6\}, \delta = 10^{-5}$). We report average accuracy and 95% confidence intervals over 10 runs. The best/second-best results are highlighted in **bold**/underline, respectively.

| DP Budget | Method | SNLI | MNLI Matched | MNLI Mismatched | SST-2 | QQP | QNLI | Average |
|---|---|---|---|---|---|---|---|---|
| $\epsilon = 6$ | FedAvg | 61.37 $_{\pm 10.26}$ | 65.45 $_{\pm 6.14}$ | 67.02 $_{\pm 5.93}$ | 89.41 $_{\pm 2.18}$ | 58.59 $_{\pm 5.27}$ | 60.70 $_{\pm 5.27}$ | 67.17 $_{\pm 2.63}$ |
| | FFA-LoRA | 62.55 $_{\pm 9.48}$ | 55.56 $_{\pm 8.58}$ | 56.39 $_{\pm 8.94}$ | 91.42 $_{\pm 0.87}$ | 64.35 $_{\pm 3.26}$ | 72.39 $_{\pm 4.96}$ | 68.02 $_{\pm 3.37}$ |
| | FLoRA | 39.14 $_{\pm 6.39}$ | 48.01 $_{\pm 10.76}$ | 48.86 $_{\pm 11.22}$ | **91.83** $_{\pm 1.13}$ | 63.18 $_{\pm 5.16}$ | 49.48 $_{\pm 0.03}$ | 59.78 $_{\pm 4.87}$ |
| | FedEX-LoRA | 54.27 $_{\pm 10.67}$ | 54.98 $_{\pm 8.16}$ | 56.02 $_{\pm 8.10}$ | 87.34 $_{\pm 1.74}$ | 53.29 $_{\pm 8.46}$ | 49.86 $_{\pm 0.35}$ | 60.86 $_{\pm 3.05}$ |
| | FedSVD (ours) | **72.77** $_{\pm 10.22}$ | **71.68** $_{\pm 3.31}$ | **73.03** $_{\pm 2.90}$ | 91.32 $_{\pm 0.85}$ | **72.42** $_{\pm 2.36}$ | **75.50** $_{\pm 4.20}$ | **76.79** $_{\pm 1.81}$ |
| $\epsilon = 3$ | FedAvg | 36.70 $_{\pm 4.56}$ | 49.91 $_{\pm 12.16}$ | 50.53 $_{\pm 12.18}$ | 61.87 $_{\pm 9.08}$ | 55.27 $_{\pm 7.89}$ | 50.00 $_{\pm 0.35}$ | 50.71 $_{\pm 4.30}$ |
| | FFA-LoRA | 56.96 $_{\pm 8.96}$ | 57.76 $_{\pm 7.09}$ | 59.19 $_{\pm 7.02}$ | **91.08** $_{\pm 1.23}$ | 68.68 $_{\pm 4.34}$ | 62.35 $_{\pm 8.07}$ | 66.00 $_{\pm 3.12}$ |
| | FLoRA | 33.42 $_{\pm 0.77}$ | 41.36 $_{\pm 13.96}$ | 41.81 $_{\pm 14.55}$ | 90.46 $_{\pm 1.86}$ | 57.91 $_{\pm 11.12}$ | 49.68 $_{\pm 0.46}$ | 52.44 $_{\pm 3.17}$ |
| | FedEX-LoRA | 55.62 $_{\pm 9.85}$ | 40.92 $_{\pm 7.12}$ | 41.32 $_{\pm 7.54}$ | 74.29 $_{\pm 8.09}$ | 50.00 $_{\pm 8.61}$ | 49.78 $_{\pm 0.32}$ | 49.28 $_{\pm 2.81}$ |
| | FedSVD (ours) | **70.89** $_{\pm 8.52}$ | **70.65** $_{\pm 3.97}$ | **72.02** $_{\pm 3.96}$ | 90.46 $_{\pm 0.66}$ | **72.65** $_{\pm 2.60}$ | **77.10** $_{\pm 1.60}$ | **75.63** $_{\pm 1.92}$ |

Figure 3: Accuracy vs. communication rounds **with DP** ($\epsilon = 6, \delta = 10^{-5}$) across 5 GLUE tasks. Curves show average accuracy over 10 runs, with shaded regions indicating 95% confidence intervals.

gradient norm of $B$ (*cf.* Eq. 8), making gradient clipping more robust under DP-SGD settings. Fig. 3 demonstrates the effectiveness of SVD re-initialization: FedSVD consistently exhibits better convergence behavior compared to FFA-LoRA across most training rounds. Although we observe a slight accuracy drop on SST-2 after round 80, FedSVD maintains strong overall accuracy, which demonstrates its robustness to DP noise and suitability for real-world federated learning deployments.

**Results on HellaSwag.** To verify the scalability of FedSVD to more complex tasks, we compare it with FedAvg and FFA-LoRA using the **HellaSwag** [39] dataset. We partition the training split with $\alpha = 0.5$ based on the activity_label field (*i.e.*, labels associated with each caption), since it does not contain explicit labels. We fine-tune SmolLM-360M [2] with LoRA under DP constraints ($\epsilon = 6, \delta = 10^{-5}$). We use the same experimental setups as in Sec. 4.1. The models are trained with the *next-token prediction* objective only on the correct endings. At test, we select the endings with the highest normalized log-likelihood:

$$\arg\max_{\mathbf{x} \in \mathcal{X}^{(c)}} \frac{1}{|\mathbf{x}|} \log p_\theta(\mathbf{x} \mid \mathbf{c}), \tag{13}$$

where $\mathbf{x}$ and $\mathbf{c}$ are the token sequences of endings and ctx, and $\mathcal{X}^{(c)}$ is the set of candidate endings. Table 3 presents results averaged over 5 runs, where FedSVD *outperforms* all baselines (**+1.34 pp**), demonstrating its effectiveness on a more complex commonsense reasoning task.

Table 3: **Results on the HellaSwag** [39] dataset.

| Method | Accuracy |
|---|---|
| FedAvg | 48.81 $_{\pm 0.28}$ |
| FFA-LoRA | 49.76 $_{\pm 0.09}$ |
| FedSVD | **51.10** $_{\pm 0.16}$ |

**Integration with DoRA.** We also show that FedSVD can be successfully integrated with **DoRA** [19]; see Table 11 in Appendix D for details.

### 4.3 Analysis

**Initialization of $A$.** To better understand the effect of initialization strategies for matrix $A$, we compare three classes of configurations. First, we randomly initialize $A$ with orthonormal rows and keep it fixed during training ($A_{\text{fixed}}$ **w/ random orthonormal**). Second, following PiSSA [23], we factorize the frozen pre-trained matrix using SVD: $W_0 = U_0 \Sigma_0 V_0^\top$ and initialize $A$ and $B$ with $\sqrt{\Sigma_0[:r,:r]}V_0^\top[:r,:]$ and $U_0[:,:r]\sqrt{\Sigma_0[:r,:r]}$, respectively. The base matrix $W_0$ is re-initialized with its residual component $W_0' = U_0[:,r+1:]\Sigma_0[r+1:,r+1:]V_0^\top[r+1:,:]$. Both $W_0'$ and $A$ are frozen, while only $B$ is updated during training ($A_{\text{fixed}}$ **w/ PiSSA**). Lastly, we consider an alternative SVD-based initialization where $A$ is periodically re-initialized with $\sqrt{\Sigma[:r,:r]}V^\top[:r,:]$ and $B$ with $U[:,:r]\sqrt{\Sigma[:r,:r]}$ from the SVD of $BA$, which does not preserve the orthonormality of $A$'s rows (FedSVD **w/o orthonormal**).

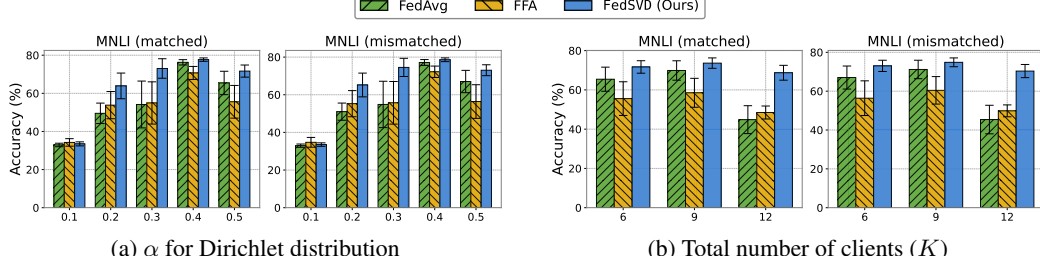

(a) $\alpha$ for Dirichlet distribution          (b) Total number of clients ($K$)

Figure 4: **(a):** Results of **varying $\alpha \in \{0.1, 0.2, 0.3, 0.4, 0.5\}$** for a Dirichlet distribution on the MNLI dataset. **(b):** Results of varying the total **number of clients** ($K \in \{6, 9, 12\}$, and $K' = 3$) on the MNLI dataset.

Table 4 shows that introducing structural priors into matrix $A$, *i.e.*, $A_{\text{fixed}}$ w/ random orthonormal, or $A_{\text{fixed}}$ w/ PiSSA, helps to stabilize training and yields better performance compared to the unstructured baseline, *i.e.*, $A_{\text{fixed}}$ (FFA-LoRA). However, when $A$ is kept fixed throughout training (methods without †), the improvements are limited, suggesting that **adaptivity plays a crucial role** beyond the structural prior itself. In addition, we find that removing the orthonormal constraint from FedSVD (denoted as

Table 4: Results on the MNLI dataset with **different initializations of $A$** under DP-SGD ($\epsilon = 6, \delta = 10^{-5}$). † indicates that $A$ matrices are periodically updated.

| Method | Matched | Mismatched |
|---|---|---|
| $A_{\text{fixed}}$ (FFA-LoRA) | $55.56 \pm 8.58$ | $56.39 \pm 8.94$ |
| $A_{\text{fixed}}$ w/ random orthonormal | $55.58 \pm 5.97$ | $56.96 \pm 5.98$ |
| $A_{\text{fixed}}$ w/ PiSSA | $66.32 \pm 2.87$ | $67.57 \pm 2.79$ |
| FedSVD w/o orthonormal† | $\underline{70.76} \pm 3.75$ | $\underline{71.86} \pm 3.79$ |
| FedSVD† (Ours) | $\mathbf{71.68} \pm 3.31$ | $\mathbf{73.03} \pm 2.90$ |

FedSVD w/o orthonormal†) degrades performance, indicating that the orthonormal structure of $A$ is not only beneficial for initialization but remains important throughout training. Although the effectiveness of enforcing the orthonormal constraint appears marginal on the MNLI dataset in Table 4 (*e.g.*, +0.92/+1.17 pp for Matched/Mismatched), it yields a much larger improvement on the SNLI dataset (*i.e.*, **+11.68 pp**), as shown in Table 10 of Appendix D.

**Heterogeneity of the data distribution ($\alpha$).** To assess the robustness of FedSVD under varying degrees of non-i.i.d. data, we partition the MNLI dataset across clients using various concentration parameters $\alpha \in \{0.1, 0.2, \ldots, 0.5\}$ for the Dirichlet distribution. For each setting, we train models under DP-SGD ($\epsilon = 6, \delta = 10^{-5}$) and report the mean and standard deviation over 5 independent runs. We compare FedSVD with FedAvg and FFA-LoRA across all levels of heterogeneity. As shown in Fig. 4a, our proposed FedSVD consistently outperforms the baselines across all tested levels of data heterogeneity, except at $\alpha = 0.1$, where extreme heterogeneity causes all methods to fail.

**Varying the number of clients ($K$).** To evaluate the robustness of each method in more realistic federated settings, we vary the total number of clients $K \in \{6, 9, 12\}$ while keeping the number of participating clients per round fixed at $K' = 3$. Here, $K = 12$ is near the *maximum feasible value*, as some clients already have fewer training samples than the data processed per round (*i.e.*, batch size×local steps $\tau$). We compare the performance of FedAvg, FFA-LoRA, and FedSVD under DP-SGD ($\epsilon = 6, \delta = 10^{-5}$) on the MNLI dataset and report the mean and standard deviation over 5 independent runs for each configuration. Fig. 4b shows that FedSVD consistently outperforms the baselines across all values of $K$. Notably, the performance degradation with increasing $K$ is significantly smaller for FedSVD, showing its robustness to the number of clients.

**Overcoming the SVD bottleneck: (1) Low-rank SVD.** Although FedSVD introduces additional overhead due to the SVD step, this cost can be substantially mitigated by using randomized low-rank approximation techniques, *e.g.*, the algorithm proposed by Halko et al. [13, Algorithm 5.1 on p. 29]. It iteratively approximates the leading singular components with high fidelity while significantly reducing computational complexity, making FedSVD feasible for practical use in large-scale federated settings. We set the number of iterations for the low-rank approximation (niter) to 2 or 10. In Table 5, the approximation (denoted as Low-rank SVD) achieves *similar accuracy* to Full SVD (*even better* with niter=2 and 10 on SNLI/MNLI-Matched and QNLI, respectively), while running approximately **60×** (niter=2) or **10×** (niter=10) **faster** than Full SVD. This demonstrates that Low-rank SVD can serve as an *efficient alternative* to full SVD without sacrificing accuracy.

**Overcoming the SVD bottleneck: (2) Frequency of SVD.** To further alleviate the computational burden, we explore reducing the frequency of SVD itself. Specifically, we conduct an ablation study in which SVD-based re-initialization is applied every 1, 2, 5, or 10 communication rounds.

Table 5: Results of `FedSVD` using the **low-rank approximation** (`Low-rank SVD`) with `niter`=2 or 10 under DP-SGD ($\epsilon = 6, \delta = 10^{-5}$). The average run-time per SVD step (in seconds) for `Full SVD` is **9.12**$_{\pm\,0.08}$, and for `Low-rank SVD` it is **0.15**$_{\pm\,0.04}$ (`niter`=2; **60× faster**) or **0.89**$_{\pm\,0.07}$ (`niter`=10; **10× faster**).

| SVD strategy | SNLI (niter=2) | MNLI (niter=2) Matched | MNLI (niter=2) Mismatched | SST-2 (niter=2) | QQP (niter=2) | QNLI (niter=2) | QNLI (niter=10) |
|---|---|---|---|---|---|---|---|
| `Full SVD` | 72.71 $_{\pm\,7.83}$ | 71.57 $_{\pm\,3.18}$ | **73.03** $_{\pm\,2.89}$ | 91.32 $_{\pm\,0.53}$ | 72.42 $_{\pm\,2.36}$ | **75.50** $_{\pm\,4.20}$ | 75.50 $_{\pm\,4.20}$ |
| `Low-rank SVD` | **74.92** $_{\pm\,5.06}$ | **72.76** $_{\pm\,1.82}$ | 72.74 $_{\pm\,1.64}$ | **92.34** $_{\pm\,0.60}$ | **76.66** $_{\pm\,1.02}$ | 68.76 $_{\pm\,7.89}$ | **79.84** $_{\pm\,2.06}$ |

Each configuration is denoted as `FedSVD` ($n$), where $n \in \{1, 2, 5, 10\}$ denotes the re-initialization interval in rounds. As shown in Fig. 5, all `FedSVD` ($n$) variants exhibit better convergence than FFA-LoRA, confirming the benefit of SVD re-initialization and its robustness to the choice of interval $n$. Given their comparable performance, variants with less frequent re-initialization offer a *favorable trade-off* when computational efficiency is prioritized. Accuracy remains *stable* across different re-initialization schedules, demonstrating the robustness of `FedSVD` to hyperparameter $n$.

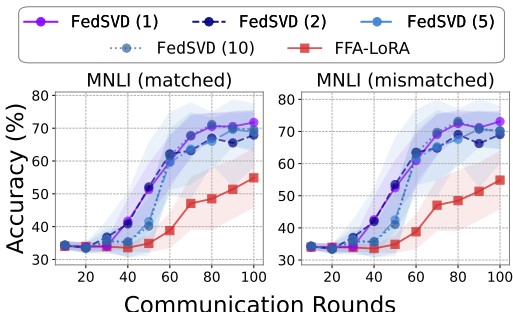

Figure 5: Results of **varying the SVD frequency** using the MNLI dataset under DP-SGD ($\epsilon = 6, \delta = 10^{-5}$).

## 5 Conclusion

In this work, we proposed `FedSVD`, a *simple yet effective* method for fine-tuning language models with DP-SGD in FL. Instead of using a fixed random matrix $A$ for LoRA, we periodically refactor the product of two LoRA adapter matrices $BA$ with SVD and initialize $A$ with the right singular vectors of $BA$. As $A$ remains untrained and SVD is applied post-privatization of $B$, our method preserves differential privacy without incurring additional noise from matrix multiplication. Empirically, `FedSVD` consistently outperforms the relevant baselines, often achieving faster convergence.

**Limitations.** Although our approach shows promising results in both private and non-private federated learning settings, the *computation of SVD* incurs additional overhead on the server side. However, since SVD is performed on low-rank matrices, this overhead can be significantly reduced by employing randomized low-rank approximation methods, such as the algorithm proposed by Halko et al. [13, Algorithm 5.1], as shown in Table 5. Another limitation is the additional communication overhead associated with the *broadcast* of the newly initialized $\hat{A}$ matrix to clients after each SVD step. However, this cost can be avoided by decentralizing the SVD computation. After aggregating $B_i$, the server computes $\hat{A}_i$ via SVD on the product $B_i \hat{A}_{i-1}$ and transmits only $B_i$ to the clients. Each client then reconstructs $\hat{A}_i$ locally using the same procedure and obtains the updated pair $(\hat{B}_i, \hat{A}_i)$. Since only $\hat{B}_i$ is optimized during training while $\hat{A}_i$ remains fixed, it is not necessary to transmit or aggregate $\hat{A}_i$ at the server. Table 6 compares the communication cost when both the server and the clients perform SVD computations, showing that our proposed `FedSVD`, along with FFA-LoRA, achieves the lowest communication cost, since only the LoRA $B$ matrix is transmitted.

Table 6: **Communication cost per round** (*i.e.*, the number of parameters exchanged between the server and clients) when using RoBERTa-large and applying LoRA with rank $r = 8$.

| Method | Comm. Cost (# parameters.) |
|---|---|
| FedAvg | 786,432 |
| FFA-LoRA | 393,216 |
| FLoRA | 52,169,730 |
| FedEX-LoRA | 52,169,730 |
| `FedSVD` (ours) | 393,216 |

**Future work.** Since our method is compatible with any FL setup employing LoRA, extending the empirical evaluation of `FedSVD` to a wider range of foundation models across different modalities is a promising direction for future work. Furthermore, a deeper theoretical analysis of `FedSVD`'s convergence dynamics, particularly for complex non-linear models, could provide valuable insights.

**Broader impact.** `FedSVD` advances data privacy in AI development by enabling stable and effective training of neural networks under differential privacy within a federated learning framework, ensuring that sensitive data remains locally available to each client. By improving the robustness of privacy-preserving fine-tuning for foundation models, `FedSVD` contributes to reducing the risk of information leakage and supports the responsible deployment of AI systems in sensitive domains.

## Acknowledgments

We express our sincere gratitude to the anonymous reviewers (**z29n**, **toPF**, **yjb1**, and **w3wU**) for their valuable feedback and efforts in helping us improve this paper.

**Funding.** This work was supported by Institute for Information & communications Technology Planning & Evaluation (IITP) grant funded by the Korea government (MSIT) (RS-2019-II190075, Artificial Intelligence Graduate School Program (KAIST), No.RS-2022-II220713, Meta-learning Applicable to Real-world Problems, No. RS-2020II200153, Penetration Security Testing of ML Model Vulnerabilities and Defense), National Research Foundation of Korea (NRF) grant funded by MSIT (No. RS-2023-00256259), a grant of the Korea Machine Learning Ledger Orchestration for Drug Discovery Project (K-MELLODDY), the Ministry of Health & Welfare and Ministry of Science and ICT, Republic of Korea (grant number: RS-2024-00460870), the Bavarian State Ministry of Science and the Arts (grant number: H.2-F1116.NÜ/61/2), and Samsung Research.

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

## Appendix

## A    Proof of Proposition 3.2

*Proof.* Let $\mathbf{x}_i \in \mathbb{R}^{d_x}$ with one-hot $\mathbf{y}_i \in \mathbb{R}^C$. Parameters:

$$A \in \mathbb{R}^{r \times d_x}, \quad B \in \mathbb{R}^{d_h \times r}, \quad W_1 \in \mathbb{R}^{d_h \times d_x}, \quad W_2 \in \mathbb{R}^{C \times d_h}.$$

Now we assume $A$ has full row rank, *i.e.*, $\operatorname{rank}(A) = r$. We define activations with forward pass as:

$$\mathbf{h}_i = (W_1 + BA)\mathbf{x}_i \in \mathbb{R}^{d_h}, \qquad \mathbf{a}_i = \operatorname{ReLU}(\mathbf{h}_i) \in \mathbb{R}^{d_h}, \qquad \mathbf{z}_i = W_2 \mathbf{a}_i \in \mathbb{R}^C.$$

With elementwise logarithm, let

$$\mathbf{p}_i = \operatorname{softmax}(\mathbf{z}_i), \qquad \ell_i = -\mathbf{y}_i^\top \log \mathbf{p}_i, \qquad \mathcal{L} = \frac{1}{n} \sum_{i=1}^{n} \ell_i.$$

Standard logit-space derivatives are

$$\frac{\partial \ell_i}{\partial z_{i,c}} = p_{i,c} - y_{i,c}, \qquad \frac{\partial^2 \ell_i}{\partial z_{i,c} \, \partial z_{i,c'}} = S_{i,cc'}, \quad S_i = \operatorname{diag}(\mathbf{p}_i) - \mathbf{p}_i \mathbf{p}_i^\top \succeq 0.$$

Let $D_i = \text{diag}(\mathbb{1}\{\mathbf{h}_i > 0\}) \in \mathbb{R}^{d_h \times d_h}$, so $D_i^2 = D_i = D_i^\top$. On any open region where the sign pattern of $\mathbf{h}_i$ is fixed, $D_i$ is constant and
$$\mathbf{a}_i = D_i\,(W_1 + BA)\,\mathbf{x}_i, \qquad \mathbf{z}_i = W_2 D_i(W_1\mathbf{x}_i) + W_2 D_i\,B\,(A\mathbf{x}_i).$$
Set $\mathbf{t}_i = (t_{i,1}, \ldots, t_{i,r}) := A\mathbf{x}_i \in \mathbb{R}^r$. Then for each $c \in \{1, \ldots, C\}$,
$$z_{i,c} = \sum_{a=1}^{d_h} W_{2,ca} D_{i,aa}(W_1\mathbf{x}_i)_a + \sum_{a=1}^{d_h} \sum_{b=1}^{r} W_{2,ca} D_{i,aa} B_{ab} t_{i,b}.$$
Hence, on a fixed mask,
$$\frac{\partial z_{i,c}}{\partial B_{ab}} = W_{2,ca} D_{i,aa} t_{i,b} \quad \text{(independent of } B\text{)}, \qquad \frac{\partial^2 z_{i,c}}{\partial B_{ab}\,\partial B_{pq}} = 0.$$
Now apply the full second-order chain rule for $\ell_i(z_i(B))$:
$$\frac{\partial^2 \ell_i}{\partial B_{ab}\,\partial B_{pq}} = \sum_{c=1}^{C} \sum_{c'=1}^{C} \frac{\partial z_{i,c}}{\partial B_{ab}} S_{i,cc'} \frac{\partial z_{i,c'}}{\partial B_{pq}} + \sum_{c=1}^{C} \frac{\partial \ell_i}{\partial z_{i,c}} \frac{\partial^2 z_{i,c}}{\partial B_{ab}\,\partial B_{pq}}.$$
The second sum vanishes (affine logits in $B$). Substituting the first derivatives gives
$$\frac{\partial^2 \ell_i}{\partial B_{ab}\,\partial B_{pq}} = t_{i,b}\, t_{i,q} \sum_{c,c'}(W_{2,ca} D_{i,aa}) S_{i,cc'} (W_{2,c'p} D_{i,pp}) = (\mathbf{t}_i\mathbf{t}_i^\top)_{bq}\, [\, D_i W_2^\top S_i W_2 D_i\,]_{ap}.$$
Therefore the per-sample Hessian (indexed by $(a,b)$ rows/cols) is
$$H_i = \nabla^2_{\text{vec}(B)}\ell_i = (\mathbf{t}_i\mathbf{t}_i^\top) \otimes \left(D_i W_2^\top S_i W_2 D_i\right) \succeq 0.$$
Averaging,
$$H = \nabla^2_{\text{vec}(B)}\mathcal{L} = \frac{1}{n}\sum_{i=1}^{n}(\mathbf{t}_i\mathbf{t}_i^\top) \otimes \left(D_i W_2^\top S_i W_2 D_i\right).$$
Let $\mathcal{A} := A \otimes I_{d_h} \in \mathbb{R}^{(r d_h)\times(d_x d_h)}$ and define
$$\mathcal{M} := \frac{1}{n}\sum_{i=1}^{n}(I_{d_h} \otimes \mathbf{x}_i)\left(D_i W_2^\top S_i W_2 D_i\right)(I_{d_h} \otimes \mathbf{x}_i^\top) \in \mathbb{R}^{(d_h d_x)\times(d_h d_x)}.$$
Then $H_k(B; A) = \mathcal{A}\mathcal{M}\mathcal{A}^\top$. Now our goal is to bound the following quantity
$$\kappa_2\big(H_k(B; A)\big) = \frac{\lambda_{\max}\big(H_k(B; A)\big)}{\lambda_{\min}(H_k(B; A))} = \frac{\lambda_{\max}(\mathcal{A}\mathcal{M}_k\mathcal{A}^\top)}{\lambda_{\min}(\mathcal{A}\mathcal{M}_k\mathcal{A}^\top)}. \tag{14}$$

Using
$$\lambda_{\max}\big(H_k(B; A)\big) = \big\|\mathcal{A}\mathcal{M}_k\mathcal{A}^\top\big\|_2 \leqslant \|\mathcal{A}\|_2^2\,\|\mathcal{M}_k\|_2 = \|\mathcal{A}\|_2^2\,\lambda_{\max}(\mathcal{M}_k)$$
and $\|\mathcal{A}\|_2 = \|I_C \otimes A\|_2 = \|A\|_2 = \sigma_{\max}(A)$, we can bound the numerator in Eq. 14 as follows:
$$\lambda_{\max}(H_k(B; A)) \leqslant \sigma_{\max}(A)^2 \lambda_{\max}(\mathcal{M}_k).$$

Let $\mathcal{R}(\mathcal{A}^\top) := \{\mathcal{A}^\top\mathbf{v} : \mathbf{v} \in \mathbb{R}^{d_h r}\}$ be an image of $\mathcal{A}^\top$. By Rayleigh quotient characterization,
$$\lambda_{\min}(\mathcal{M}_k|_{\mathcal{R}(\mathcal{A}^\top)}) = \min_{\mathbf{w}\in\mathcal{R}(\mathcal{A}^\top),\mathbf{w}\neq\mathbf{0}} \frac{\mathbf{w}^\top\mathcal{M}_k\mathbf{w}}{\mathbf{w}^\top\mathbf{w}} = \min_{\mathbf{w}\in\mathcal{R}(\mathcal{A}^\top),\|\mathbf{w}\|_2=1} \mathbf{w}^\top\mathcal{M}_k\mathbf{w},$$
for every $\mathbf{w} \in \mathcal{R}(\mathcal{A}^\top)$ we have
$$\mathbf{w}^\top\mathcal{M}_k\mathbf{w} \geqslant \lambda_{\min}(\mathcal{M}_k|_{\mathcal{R}(\mathcal{A}^\top)})\,\|\mathbf{w}\|_2^2.$$
Applying this to $\mathbf{w} = \mathcal{A}^\top\mathbf{v}$ with $\mathbf{v} \in \mathbb{R}^{d_h r}$ and then minimizing over $\|\mathbf{v}\|_2 = 1$ gives
$$\begin{aligned}
\lambda_{\min}(H_k(B; A)) &= \min_{\|\mathbf{v}\|_2=1} \mathbf{v}^\top\mathcal{A}\mathcal{M}_k\mathcal{A}^\top\mathbf{v} \\
&= \min_{\|\mathbf{v}\|_2=1} \mathbf{w}^\top\mathcal{M}_k\mathbf{w} \\
&\geqslant \lambda_{\min}(\mathcal{M}_k|_{\mathcal{R}(\mathcal{A}^\top)}) \min_{\|\mathbf{v}\|_2=1} \big\|\mathcal{A}^\top\mathbf{v}\big\|_2^2.
\end{aligned}$$

Using Rayleigh quotient characterization again,

$$\min_{\|\mathbf{v}\|_2 = 1} \mathbf{v}^\top \mathcal{A}\mathcal{A}^\top \mathbf{v} = \lambda_{\min}(\mathcal{A}\mathcal{A}^\top) = \sigma_{\min}^2(\mathcal{A}) = \sigma_{\min}^2(A).$$

Thus we obtain

$$\kappa_2\left(H_k(B;A)\right) \leqslant \frac{\sigma_{\max}(A)^2 \lambda_{\max}(\mathcal{M}_k)}{\sigma_{\min}(A)^2 \lambda_{\min}(\mathcal{M}_k|_{\mathcal{R}(\mathcal{A}^\top)})} = \kappa_2(A)^2 \frac{\lambda_{\max}(\mathcal{M}_k)}{\lambda_{\min}(\mathcal{M}_k|_{\mathcal{R}(\mathcal{A}^\top)})}. \tag{15}$$

If the rows of $A$ are orthonormal (so $AA^\top = I_r$ and $\sigma_{\max}(A) = \sigma_{\min}(A) = 1$), then

$$\kappa_2\left(H_k(B;A)\right) \leqslant \frac{\lambda_{\max}(\mathcal{M}_k)}{\lambda_{\min}(\mathcal{M}_k|_{\mathcal{R}(\mathcal{A}^\top)})}. \tag{16}$$

$\square$

## B  Related Work

**Federated learning.** Federated Learning (FL) enables decentralized clients to collaboratively train models without sharing raw data. FedAvg [21] averages locally updated model weights to form a global model, offering a simple yet effective baseline. Built upon FedAvg, recent work has explored integrating Low-Rank Adaptation [LoRA; 16] into FL to reduce communication and computation overhead during model fine-tuning. For instance, Fed-IT [40] updates the adapter matrices $A$ and $B$ of LoRA, averages each matrices separately. To aggregate product of $B$ and $A$, several methods have been proposed. FedEx-LoRA [28] introduces an additional correction matrix to mitigate aggregation error. FLoRA [34] stacks adapter matrices and reinitializes them randomly at the end of each communication round. FFA-LoRA [31] proposes to use a fixed randomly initialized matrix $A$, while training and aggregating only $B$. Lastly, Fed-SA [12] proposes learning both matrices $A$ and $B$, but shares only $A$ during aggregation. Our method is based on FFA-LoRA; however, we reinitialize the adapter matrices after aggregation to promote gradient stability and learning efficacy. Instead of using a fixed random matrix for $A$, we periodically reinitialize $A$ using orthonormal bases via singular value decomposition (SVD) of $BA$, which empirically accelerates optimization.

**Differential privacy guaranteed federated fine-tuning.** $(\epsilon, \delta)$-differential privacy [DP; 9] provides a rigorous framework ensuring that models trained on neighboring datasets, differing by only one data point, produce similar outputs, thereby preserving individual privacy. DP-SGD [30, 3, 1] brings this guarantee to deep learning by adding noise to stochastic gradient updates. In FL, privacy guarantees depend on whether the central server is trusted. In the centralized DP setting, clients send raw updates without local privacy, and DP is applied during global aggregation [22]. In the local DP setting, which assumes an untrusted server, each client ensures its update is differentially private before communication [36, 18, 25]. Our work adopts this stronger setting: we apply DP at the client level, so any shared updates (*i.e.*, model parameters) are already privatized. By the composition property of DP, the final global model also satisfies DP. DP-SGD is unstable with large numbers of trainable parameters due to increased gradient sensitivity and noise injection [1, 38]. To address this, FFA-LoRA [31] fixes the adapter matrix $A$ in LoRA to reduce trainable parameters, limiting noise amplification and avoiding quadratic noise growth.

**Parameter efficient fine-tuning.** To mitigate the computational cost of fine-tuning language models, LoRA [16] injects trainable low-rank adapter matrices into some of model components. Subsequent works have proposed variants to improve adaptability and efficiency. For example, DeltaLoRA [42] improves LoRA's expressivity by combining original weights with adapter outputs, thereby enhancing the representational power. LoSparse [17] integrates LoRA with sparsity constraints to prevent the pruning of essential neurons. DoRA [19] separates the magnitude and direction of the update by learning a scaling factor for the update $\Delta W$, while keeping the direction determined by the LoRA update $BA$.

Unlike these approaches, which aim to learn expressive low-rank approximations of weight updates, PiSSA [23] takes a more structural approach. It first decomposes the original weight matrix using SVD, then fine-tunes only the low-rank components corresponding to the top-$r$ singular values, while freezing the residual parts. Our method differs from PiSSA in two key aspects. First, rather than decomposing the pretrained weights, we perform SVD on the aggregated adapter product $BA$

to reinitialize low-rank components after aggregation of optimized $B$ on the client side. This is distinct from PiSSA's fixed decomposition of model weights. Second, we enforce the rows of $A$ to be orthonormal by initializing them with right singular vectors of $BA$, which empirically stabilizes training and accelerates optimization compared to a non-orthonormal structure. AdaLoRA [41] dynamically learns the optimal rank by parameterizing incremental updates through an SVD to dynamically prune and reallocate a rank budget across layers based on the magnitude of the singular values during training. Unlike AdaLoRA, we employ SVD to refactor the aggregated adapter product $BA$ and enforce the rows of $A$ to be orthonormal by initializing them with right singular vectors.

## C  Dataset Statistics

In this section, we summarize the statistics of the datasets used in our experiments (Table 7), and present the per-label data distribution across clients (%) for both two-class (SST-2, QQP, QNLI) and three-class (MNLI, SNLI) datasets using a Dirichlet distribution with $\alpha = 0.5$ and six clients in total (Table 8).

Table 7: An **overview of datasets** used in our experiments.

| Dataset | # Classes | # Train | # Val | # Test |
|---|---|---|---|---|
| SNLI | 3 | 550,152 | 10,000 | 10,000 |
| MNLI (matched) | 3 | 392,702 | 9,815 | - |
| MNLI (mismatched) | | | 9,832 | - |
| SST-2 | 2 | 67,349 | 872 | - |
| QQP | 2 | 363,846 | 40,430 | - |
| QNLI | 2 | 104,743 | 5,463 | - |
| HellaSwag | N/A | 39,905 | 10,042 | - |

Table 8: **Per-label data distribution** across clients (%) for datasets with two labels (SST-2, QQP, QNLI) and three labels (MNLI, SNLI) under the Dirichlet partition ($\alpha = 0.5$, 6 clients). For the HellaSwag dataset, which contains 137 distinct `activity_labels`, we do not report the detailed per-label distribution here; however, the partitioning strategy can be found at `https://github.com/seanie12/fed-svd`.

| # Labels | Label | Client 0 | Client 1 | Client 2 | Client 3 | Client 4 | Client 5 |
|---|---|---|---|---|---|---|---|
| 2 | 0 | 0.196 | **0.469** | 0.187 | 0.018 | 0.130 | 0.001 |
| | 1 | 0.020 | 0.100 | 0.004 | 0.089 | 0.186 | **0.600** |
| 3 | 0 | 0.104 | 0.025 | **0.673** | 0.024 | 0.100 | 0.073 |
| | 1 | 0.049 | 0.016 | 0.007 | 0.405 | 0.058 | **0.465** |
| | 2 | **0.333** | 0.168 | 0.113 | 0.000 | 0.064 | 0.322 |

## D  Additional Experiments

In this section, we present additional experiments to empirically support the effectiveness of the proposed FedSVD.

Table 9: **Comparison on condition numbers**, $\kappa_2(H(B; A)) = \lambda_{\max}(H(B; A))/\lambda_{\min}(H(B; A))$.

| Method | 0 | 1k | 2k | 3k | 4k | 5k |
|---|---|---|---|---|---|---|
| FFA-LoRA | 10.18 | 10.15 | 9.78 | 10.09 | 10.13 | 10.23 |
| FedSVD | 1.67 | 1.52 | 1.51 | 1.50 | 1.50 | 1.51 |
| Oracle | 1.06 | 1.01 | 1.02 | 1.04 | 1.03 | 1.02 |

**Empirical validation of Proposition 3.2.**  To empirically support Proposition 3.2, we consider a simple logistic regression setup which allows us to directly compute the **actual condition number** during optimization. Therefore, we directly measure the condition number $\kappa_2(H(B; A)) =$

$\lambda_{\max}(H(B;A))/\lambda_{\min}(H(B;A))$ for FFA-LoRA, FedSVD, and oracle. Specifically, using the SST-2 dataset, we (1) extract features $\mathbf{X} \in \mathbb{R}^{n \times d_{\text{in}}}$ using a pretrained BERT model, (2) train a logistic regression head on top of these frozen features, and (3) compute the Hessian's condition number directly during optimization. As shown in Table 9, we observe that the condition number of FedSVD remains consistently smaller than that of FFA-LoRA throughout training (up to 5,000 iterations), closely matching the oracle. This confirms the practical advantage of FedSVD for optimization.

Table 10: Results on the SNLI [5] dataset with **different initializations of $A$**. We report average accuracy and 95% confidence intervals over 5 runs.

| Method | Accuracy |
|---|---|
| FedSVD w/o orthonormal | $69.48_{\pm 9.45}$ |
| FedSVD | $\mathbf{81.16}_{\pm 2.37}$ |

**Impact of orthonormal initialization on the SNLI dataset.** In Table 4, we present the impact of different initializations of $A$ and the effect of orthonormality. The improvement from enforcing orthonormality (row 4: FedSVD w/o orthonormal vs. row 5: FedSVD (Ours)) appears marginal on the MNLI dataset—*e.g.*, +0.92 percentage points (pp) for Matched and +1.17 pp for Mismatched. However, we find that the influence of orthonormality can vary considerably across datasets. To further investigate this, we conducted an additional experiment on the SNLI dataset. As shown in Table 10, maintaining the orthonormal structure yields a substantial performance gain of **nearly 12 pp**.

**Integration of FedSVD to DoRA [19].** To investigate the potential of FedSVD on different parameter-efficient fine-tuning methods, we conduct experiments using DoRA by learning only the $B \in \mathbb{R}^{d_{\text{out}} \times r}$ matrix and freezing the $A \in \mathbb{R}^{r \times d_{\text{in}}}$ matrix, the magnitude vector $\mathbf{m} \in \mathbb{R}^{d_{\text{out}}}$, and the initial weight $W_0 \in \mathbb{R}^{d_{\text{out}} \times d_{\text{in}}}$. For a given input $\mathbf{x} \in \mathbb{R}^{d_{\text{in}}}$, the output of the DoRA layer is defined as

$$\text{diag}(\mathbf{m}) \cdot \text{diag}(\|W + BA\|_{2,\text{row}})^{-1} \cdot (W_0 + BA)\mathbf{x},$$

where $\|W + BA\|_{2,\text{row}} \in \mathbb{R}^{d_{\text{out}}}$ is a row-wise norm. After computing the SVD of $BA$, we re-initialize the magnitude vector

$$\mathbf{m} \leftarrow \text{diag}(\mathbf{m}) \cdot \text{diag}(\|W + BA\|_{2,\text{row}})^{-1} \cdot (W_0 + BA).$$

The matrices $A$ and $B$ are re-initialized as in FedSVD with LoRA, *i.e.*, $B = U[:,:r]\Sigma[:,:r]$ and $A = V[:,:r]^\top$.

Table 11: **Results with DoRA** [19] on the MLNI dataset. We report average accuracy and 95% confidence intervals over 5 runs.

| Method | Matched | Mismatched |
|---|---|---|
| FedSVD w/ LoRA | $71.57_{\pm 3.18}$ | $73.03_{\pm 2.89}$ |
| FedSVD w/ DoRA | $\mathbf{72.12}_{\pm 2.65}$ | $\mathbf{73.13}_{\pm 2.69}$ |

In Table 11, we observe that FedSVD with DoRA shows a similar performance to FedSVD with LoRA on the MNLI dataset, demonstrating the generalizability of FedSVD across different parameter-efficient fine-tuning parameterizations. We note that DoRA is known to bring benefits primarily in complex tasks (*e.g.*, image generation, text generation), and its improvements on text classification benchmarks are often marginal (*e.g.*, Table 3 in [23]).

