# OpenReview forum: "FedSVD: Adaptive Orthogonalization for Private Federated Learning with LoRA"
_NeurIPS.cc/2025/Conference — NeurIPS 2025 poster_

### Official Review · Reviewer_w3wU · 2025-07-01

**Clarity:** 4
**Significance:** 3
**Originality:** 2
**Rating:** 5
**Confidence:** 3

**Summary:**

This paper proposes a new algorithm, called FedSVD, which allows efficient LoRa fine-tuning in the federated setting. More precisely, a difficulty in the federated setting is that local updates cannot be easily summed up while preserving the low-rank structure. One way to mitigate this, for the update rule \$W\_0 + \Delta W = W\_0 + BA\$, is to keep \$A\$ fixed over time, as was previously done by FFA-LoRA. FedSVD improves upon this existing method by proposing to update \$A\$ on the server side as well. After presenting the algorithm, the authors theoretically analyze the algorithm, deriving privacy properties and Hessian properties, and demonstrate empirically that FedSVD improves upon FFA-LoRA.

**Questions:**

- Do you think FedSVD is optimal for federated LoRA fine-tuning in terms of the utility of the final result? Is it possible to consider other update rules that would lead to better results?
- The numerical experiments only test a very small number of clients (maximum 12). Can the method scale? Did you have a specific reason to keep it very small? In particular, would the algorithm perform well if only a small portion of the clients is updated at a given round (\$K' \ll K\$)?
- I am not sure I understand the heterogeneity you studied (l.277). Can you explain your method? Could you test more diverse heterogeneity settings (covariate shift, concept drift, unbalanced clients), or explain how your method would behave?

**Ethical Concerns:**

["NO or VERY MINOR ethics concerns only"]

**Final Justification:**

The author did a very impressive rebuttal with additional experiments that addressed all the questions I could think of when writing my review. To the extent of my knowledge in the specific topic, it seems that the proposed method indeed provides significant improvement with respect to existing methods and its presentation is very clear.

**Limitations:**

yes

**Quality:**

3

**Strengths And Weaknesses:**

+ The paper is very clearly written, easy to follow, with clear explanations of the setup, the theoretical results, and the experiments.
+ The topic is very timely, as adapting federated learning to foundation models and using techniques such as low-rank updates could interest many people in the community.
+ The proposed method is simple and could be implemented in practice.

----

- The optimality of the approach is not clear; see questions.
- The experiments and the privacy part could be more developed; see questions.
- The paper might feel a bit incremental with respect to FFA-LoRA. However, I don't believe this to be a major issue, as the new algorithm still needed to be properly written up and has some theoretical results, but people more familiar with the field might disagree.

---

> ### Author Rebuttal · Authors · 2025-07-30
>
> We sincerely thank you for your time, effort, and the clarity of your feedback. We are grateful that other reviewers appreciated the **simple and effective design** (z79n, yjb1, toPF), the robust and consistent experimental results (z79n, toPF), the **clear presentation** (z79n, yjb1), the **theoretical contribution** (toPF), and the **practical relevance** (toPF, yjb1). Below, we address your concerns and would be happy to discuss further during the discussion period.
>
> >**[Q1]** The paper might feel a bit incremental with respect to FFA-LoRA. However, I don't believe this to be a major issue, as the new algorithm still needed to be properly written up and has some theoretical results, but people more familiar with the field might disagree.
> - We are very grateful to the reviewer for their positive evaluation and for the thoughtful comment regarding the paper's relationship with FFA-LoRA. This is an excellent point, and we appreciate the opportunity to clarify the key distinctions and contributions of our work.
> - We agree that $\texttt{FedSVD}$  is motivated by the same challenges identified by prior work like FFA-LoRA. Our hope with $\texttt{FedSVD}$  was to introduce a new mechanism that addresses these challenges in a more fundamental and adaptive way.
> - The key distinction of $\texttt{FedSVD}$  is its shift from FFA-LoRA's **static, random projection** to a **dynamic and adaptive** one. Instead of using a fixed $A$ matrix, $\texttt{FedSVD}$ periodically refactorizes the aggregated $BA$ update via SVD, allowing $A$ to evolve and **align with the principal directions of the client updates**.
> - This adaptive orthogonalization provides a **strong theoretical basis** for our method's success, leading to a better-conditioned optimization landscape (`Theorem 2.2`) and superior gradient signal preservation under DP-SGD (`Eq. 8`).
> - These advantages are most evident in our empirical results. In the challenging private setting, $\texttt{FedSVD}$ achieves an **8.77% improvement over FFA-LoRA** (`Table 2`), demonstrating a substantial and non-incremental gain. We believe this significant improvement, backed by our theoretical analysis, highlights that our adaptive approach is a meaningful step forward.
>
> ---
>
> >**[Q2]** Do you think $\texttt{FedSVD}$  is optimal for federated LoRA fine-tuning in terms of the utility of the final result? Is it possible to consider other update rules that would lead to better results?
> - This is an excellent question. We thank you for encouraging us to think about the broader context and future directions for this work.
> - Regarding optimality, we would position $\texttt{FedSVD}$ as a **principled and highly effective approach** rather than a universally optimal one. Its strength lies in its specific design: by using SVD to periodically reparameterize the LoRA matrices, it enforces a beneficial orthonormal structure on the A matrix without requiring it to be trained.
> - As our analysis and results show, this leads to a better-conditioned optimization landscape and superior performance, particularly under the constraints of differential privacy.
> - The question about other possible update rules points to several exciting avenues for future research. We completely agree that exploring alternatives is a promising direction.
> - One could explore combining $\texttt{FedSVD}$ with methods like **DoRA** [1], which decouples the update's magnitude and direction. Our SVD-based approach could define the orthonormal directions ($A$ matrix), while a separate mechanism could learn the optimal scaling magnitude, potentially offering another layer of adaptability.
> - Furthermore, prompted by your question and our own limitations in `L320-329`, we've already run preliminary experiments to address the computational cost of SVD. We compare our standard $\texttt{FedSVD}$ with a version using a **highly efficient low-rank SVD approximation** [2]. The results are very promising:
>
>     **[Table R1. Experiments with low-rank SVD.]**
>     |Method|Matched Acc|Mismatched Acc|Avg. Run-time|
>     |-|-|-|-|
>     |$\texttt{FedSVD}$ w/ full SVD|$71.57\pm3.18$|$73.03\pm2.89$|$9.07\text{s}\pm0.07$|
>     |$\texttt{FedSVD}$ w/ low-rank SVD|$72.76\pm1.82$|$72.74\pm1.64$|$0.13\text{s}\pm0.06$|
> - These results show we can dramatically reduce the server-side computation time by over $\times69$ (from $9.07$s to $0.13$s) while maintaining on-par accuracy. This confirms that the benefits of $\texttt{FedSVD}$  can be achieved with **significantly improved practical efficiency**, strengthening its viability for real-world deployment.
>
> ---
>
> >**[Q3]** The numerical experiments only test a very small number of clients (maximum 12). Can the method scale? Did you have a specific reason to keep it very small? In particular, would the algorithm perform well if only a small portion of the clients is updated at a given round ($K' \ll K$)?
>
> - We thank you for this very important question, which is a critical consideration for any federated learning algorithm.
> - The choice to test with up to 12 clients was primarily driven by the significant computational resources required to simulate federated fine-tuning of language models like RoBERTa-large, for which we used 3 NVIDIA RTX A6000 GPUs. However, we also specifically designed our experiments to be **more challenging than those in the baseline work** we build upon (e.g., FFA-LoRA, which used $K=3$ clients).
> - Regarding performance when only a small fraction of clients participate per round ($K'\ll K$). We designed the experiment in `Fig. 4b` of our paper to investigate this exact scenario. In that analysis: We kept the number of participating clients fixed at K′=3. We then increased the total number of clients $K$ from 6, to 9, and then to 12, which directly simulates a decreasing participation ratio.
> - The results in `Fig. 4b` show that while all methods experience some performance degradation as the client pool grows, $\texttt{FedSVD}$ 's performance degrades much less  than both FedAvg and FFA-LORA. This suggests that our adaptive orthogonalization method is indeed more robust and better suited for scalable deployments where only a small subset of clients is active in any given round.
> - Therefore, while a massive-scale experiment would be the ultimate validation, the combination of $\texttt{FedSVD}$ 's observed robustness in our tests and its inherent design for efficient, principled aggregation gives us strong reason to believe it would maintain its advantages in more drastic, large-scale settings.
> ---
>
> >**[Q4]** I am not sure I understand the heterogeneity you studied (l.277). Can you explain your method?
>
> - Thank you for this question, which gives us the opportunity to clarify our methodology and highlight the robustness of $\texttt{FedSVD}$ in more diverse settings.
>
> - To emulate non-i.i.d. data conditions with total $K$ clients, client data proportions are sampled using a Dirichlet distribution. Specifically, for each class, we sample a size‑$K$ simplex vector from a Dirichlet distribution with parameter $\alpha$ (i.e., $\text{Dir}(\alpha, \ldots, \alpha)$), which determines the proportion of that class assigned to each client. This method induces a **label distribution skew**, where each client receives a different mixture of data from the various classes.
>
> - Our results in `Fig. 4a` show that $\texttt{FedSVD}$  consistently outperforms baselines across various levels of this label skew, demonstrating its robustness to this type of heterogeneity.
>
> - We will include the clarification in the revised version of the paper.
>
> > **[Q5]** Could you test more diverse heterogeneity settings (covariate shift, concept drift, unbalanced clients), or explain how your method would behave?
>
> - Thank you for pointing this out. **Covariate shift occurs when clients have different distributions** of input data $p(x)$. Concept drift is a more challenging scenario where the **underlying relationship between inputs and outputs** $p(y∣x)$ differs between clients.
> - For both heterogeneity settings, we believe the adaptive nature of $\texttt{FedSVD}$ is crucial. By periodically re-computing the $A$ matrix, $\texttt{FedSVD}$ **can track changes and find a compromise representation** that better accommodates the different concepts present in the network. While severe either covariate or concept drift is difficult for any global model, $\texttt{FedSVD}$ 's ability to evolve its low-rank subspace makes it inherently more capable of finding a robust, generalized solution than a static method like FFA-LoRA.
> - However, we respectfully consider these two heterogeneity settings to be beyond the scope of this paper.
> - Furthermore, our paper already covers **imbalanced clients (some clients have much more data)** through the aggregation scheme. As described in `Alg. 1` and `Eq. 3`, we perform a weighted average of the client $B$ matrices, where each client's update is weighted by the size of its local dataset ($n_k$). This ensures that clients with more data have a proportionally larger influence on the global model, which is a standard and effective technique for handling data imbalance.
> - Empirically, by sampling client data proportions from a Dirichlet distribution with parameter $\alpha$ (`L218–219`), **we simulate varying degrees of client imbalance**.
> - `Fig. 4a` shows that all methods fail to effectively learn in the most severe case ($\alpha=0.2$), while $\texttt{FedSVD}$ demonstrates its robustness to unbalanced clients.
>
> ---
>
> ## References
> [1] Liu, Shih-Yang, et al. "DoRA: Weight-decomposed low-rank adaptation." ICML. 2024.
>
> [2] Halko, Nathan, Per-Gunnar Martinsson, and Joel A. Tropp. "Finding structure with randomness: Probabilistic algorithms for constructing approximate matrix decompositions." SIAM review 53.2 (2011): 217-288.

---

> > ### Comment · Reviewer_w3wU · 2025-08-01
> > **Thanks for detailed answer, interesting point 2**
> >
> > I thank the authors for their detailed answer. I find your answer in Q2 quite exciting, maybe this would be worth pursuing before publication?
> > Overall, I keep my impression that this paper has interesting ideas and promising results, but could easily get stronger, so I keep a moderate support for acceptance

---

> ### Author Response · Authors · 2025-08-06
>
> We sincerely appreciate your responses to our rebuttal and participation efforts during this discussion period. We response to your question in the below:
>
> ---
> > I thank the authors for their detailed answer. I find your answer in Q2 quite exciting, maybe this would be worth pursuing before publication? Overall, I keep my impression that this paper has interesting ideas and promising results, but could easily get stronger, so I keep a moderate support for acceptance
> - Thank you for finding our answer to **[Q2]** interesting. Our answer includes two potential extensions of $\texttt{FedSVD}$: 1) one of advanced LoRA variants, DoRA [1], and 2) highly efficient low-rank SVD approximation [2].
> - During this discussion period, we first implement an integration of $\texttt{FedSVD}$ to DoRA by learning only the $B\in\mathbb{R}^{d_\text{out}\times r}$ matrix and freezing the $A\in\mathbb{R}^{r\times d_\text{in}}$ matrix, **the magnitude vector** $\textbf{m}\in\mathbb{R}^{d_\text{out}}$, and the initial weight $W_0\in\mathbb{R}^{d_\text{out}\times d_\text{in}}$. For given an input $\mathbf{x}\in\mathbb{R}^{d_\text{in}}$, the output of DoRA layer is defined as $\text{diag}(\mathbf{m})\cdot \text{diag}(\lVert W+BA \rVert_c)^{-1}\cdot(W_0 + BA)\mathbf{x}$, where $\lVert W+BA \rVert_c \in \mathbb{R}^{d_\text{out}}$ is a row-wise norm.
> - After computing SVD of $BA$, we re-initialize the magnitude vector $\mathbf{m}\leftarrow \text{diag}(\mathbf{m})\cdot \text{diag}(\lVert W+BA \rVert_c)^{-1}\cdot(W_0 + BA)$. $A$ and $B$ are re-reinitialized as the same as $\texttt{FedSVD}$ with LoRA, i.e., $B=U[:, :r]\Sigma[:r, :r]$ and $A=V^\top[:r,:]$.
> - In `Table R2`, we observe that $\texttt{FedSVD}$ w/ DoRA [1] shows similar performance to $\texttt{FedSVD}$ w/ LoRA, demonstrating the generalizability of $\texttt{FedSVD}$ across different parameter-efficient fine-tuning parameterizations. We note that DoRA is known to bring benefits primarily in complex tasks (e.g., image generation, text generation), and its improvements on text classification benchmarks are often marginal (e.g., `Table 3` in Meng et al., 2024 [3]).
>
>     **[Table R2. Experiments with DoRA.]**
>     |Method|MNLI (Matched)|MNLI (Mismatched)|
>     |-|-|-|
>     |$\texttt{FedSVD}$ w/ LoRA|$71.57\pm3.18$|$73.03\pm2.89$|
>     |$\texttt{FedSVD}$ w/ DoRA [1]|$72.13\pm2.65$|$73.13\pm2.69$|
> - As you have suggested, we also extend the experiments on $\texttt{FedSVD}$ with highly efficient low-rank SVD approximation [2] (in `Table R1`) to other datasets in the GLUE benchmark.
>
>     **[Table R3. Experiments with low-rank SVD.]**
>     |Method|MNLI (Matched)|MNLI (Mismatched)|SST-2|QQP|QNLI|QNLI (iter=10)|Avg. Run-time|Avg. Run-time (niter=10)|
>     |-|-|-|-|-|-|-|-|-|
>     |$\texttt{FedSVD}$ w/ full SVD|$71.57\pm3.18$|$73.03\pm2.89$|$91.32\pm0.35$|$72.42\pm2.36$|$75.50\pm4.20$|$75.50\pm4.20$|$9.12\text{s}\pm0.08$|-|
>     |$\texttt{FedSVD}$ w/ low-rank SVD|$72.76\pm1.82$|$72.74\pm1.64$|$92.34\pm0.60$|$76.66\pm1.02$|$68.76\pm7.89$|$79.84\pm2.06$|$0.15\text{s}\pm0.04$|$0.89\text{s}\pm0.07$|
> - `Table R3` shows that the performance of $\texttt{FedSVD}$ w/ low-rank SVD matches that of $\texttt{FedSVD}$ w/ full SVD on the GLUE benchmark, while significantly reducing the average run-time to negligible costs. This confirms that the computational overhead of our method can be made negligible.
> - For the QNLI dataset, we observed some instability with the standard approximation settings. By increasing the number of iterations for the approximation algorithm ($2\rightarrow10$), we were able to achieve stable performance and significantly improve accuracy, as reflected in the QNLI (iter=10) column.
> - Interestingly, we also observed that the approximate SVD can yield slightly better accuracy than the full SVD. We hypothesize that this counter-intuitive result is due to an implicit regularization effect. The full SVD may perfectly capture noisy variations specific to a single round's batch of client updates. In contrast, the approximate SVD inherently smooths this decomposition, focusing only on the most robust, significant update directions. This can prevent the model from overfitting to round-specific noise, leading to a more stable and generalizable learning path.
> - Specifically, approximation algorithm from [2] begins with a random projection (often represented by $Q$ in `Algorithm 5.1`) of the input matrix. In the context of DP-SGD, where Gaussian noise is added to gradients, this initial random projection can act as an **effective noise filter**. By concentrating the SVD computation on a smaller, '**pre-filtered**' matrix, the method ensures that the extracted principal directions are less susceptible to the perturbations of differential privacy, leading to a more stable aggregation in our $\texttt{FedSVD}$ approach.
> - Therefore, these results not only validate that FedSVD can be made highly efficient but also suggest that using an SVD approximation can offer a beneficial regularizing effect.

---

> > ### Author Response · Authors · 2025-08-06
> >
> > ---
> >
> > ### References
> >
> > [1] Liu, Shih-Yang, et al. "DoRA: Weight-decomposed low-rank adaptation." ICML. 2024.
> >
> > [2] Halko, Nathan, Per-Gunnar Martinsson, and Joel A. Tropp. "Finding structure with randomness: Probabilistic algorithms for constructing approximate matrix decompositions." SIAM review 53.2 (2011): 217-288.
> >
> > [3] Meng, Fanxu, Zhaohui Wang, and Muhan Zhang. "PISSA: Principal singular values and singular vectors adaptation of large language models." NeurIPS. 2024.

---

> > > ### Author Response · Authors · 2025-08-07
> > >
> > > Dear Reviewer w3wU
> > >
> > > As the discussion period is coming to a close, if there are any remaining concerns, we would be happy to clarify them further. Otherwise, if you feel that your concerns have been sufficiently addressed, we would be grateful if this could be reflected in your final score.
> > >
> > > Best Regards,
> > >
> > > The Authors

---

> > > > ### Comment · Reviewer_w3wU · 2025-08-07
> > > >
> > > > Dear Authors,
> > > >
> > > > Thanks for your additional experiments and your explanations during the discussion phase. Your answers indeed address all the concerns I have and I hope that all the modification will be in the final version.
> > > >
> > > > Best

---

> ### Author Response · Authors · 2025-08-09
>
> Thank you very much for your thoughtful feedback and for acknowledging our additional experiments and explanations during the discussion phase. We are glad to hear that our responses addressed your concerns. We will ensure that all the discussed modifications are properly incorporated into the final version of the paper.
>
> Best regards,
>
> The Authors

---

### Official Review · Reviewer_yjb1 · 2025-07-02

**Clarity:** 3
**Significance:** 3
**Originality:** 2
**Rating:** 4
**Confidence:** 4

**Summary:**

This submission address low-rank adaptation in federated learning (FL) under the use of differential private stochastic gradient descent (DP-SGD). The authors tackle a known problem in literature, which is related to the noise amplification of DP when applied to both matrices (A,B) of the low-rank reparametrization. Similarly to previous works, they propose to explicitly train locally only the B matrix, while A ramains fixed during each round. Differently from other works, the A matrix changes across rounds, as an effect of further singular value decomposition at the server side. The authors provide a motivation for the proposed approach, highlighting the orthonormalization of A as a factor that promotes better conditioned optimization.
Experiments show promising results on 5 GLUE datasets, with/without DP.

**Questions:**

* Can the authors explain why other intuitive strategies for avoiding the noise amplification (see example above) are not equally valid ad the proposed approach?
* Can the authors better motivate why the proposed approach does not limit the expressiveness of the decomposition, even if the matrix A is effectively never directly updated by training?
* Can the authors discuss theoretically how the techniques relates to priors works and guarantess with low-rank training?

**Ethical Concerns:**

["NO or VERY MINOR ethics concerns only"]

**Final Justification:**

The authors sufficiently addressed my core concerns, so I turn my recommendations towards acceptance. The author provided detailed responses and additional experiments following the suggestions arising in our discussion.
Among those, the controlled experiments that show the lower condition number are greatly appreciated, since serve as empirical demonstration of the main theoretical claim. Other experiments provided in their last message show that simpler baselines do not achieve a comparably good result even with additional tuning, which further strenghtens the empirical part.

Those motivations and evidence do clarify theoretical motivation of the method and show why refactorization of BA product is necessary for superior performance, and for this reason I am revising my score from 2 to 4.

There are mangin for improvement in clarity (I trust authors will enhance incorporating suggestions) and in proposing controlled experiments directly tied to measure the distance from the proposed solution (i.e. refactorize BA via SVD) and what should be optimal (but infeasible in practice in FL) one.

**Limitations:**

Yes, even if the limitations appear significant for large language models (e.g. computing SVD very often)

**Quality:**

2

**Strengths And Weaknesses:**

## Strenghts

* **The paper is well written:** the paper is easy to follow, sections are clear and reasonable coverage of closely related approaches is provided. Some details appear superflous for a research paper, and risk to be not accurate, such as eq. 1, which is just the GD update rule at each local step and the def. of the cross-entropy loss. This could be a typo, since the algorithms should use SGD, and not GD locally (as per eq. 1, the loss is calculated on the entire local dataset $\mathcal{D}_k$).
* **The proposed approach is simple:** conceptually, the approach is of straightforward understanding, and it is trivial to implement. Under full comprehension of the induced dynamics, which is not trivial at all (see weaknesses on that), the approach could have immediate practical implementation.

## Weaknesses
1. **Weak motivation for the proposed approach:** the aim of the work is to avoid the noise amplification that follows the use of DP-SGD in FL. To do so, authors rely on what other works do, that is train locally only the B matrix, and add on top a further SVD at the server after aggregation with the purpose of not limiting the expressiveness of A. However, the motivation behind the precise choice of recomputing an SVD after training only B is not clear, it seems an heuristic similar to other intuitive ideas one can come up. For example, clients could optimize A or B depending on the round: this would similarly avoid the noise amplification and effectively train both matrices, with the same communication complexity and without the burden of the additional SVD. Without stronger motivations for the precise steps of the proposed algorithm, this work does not offer a clear advancement of our understanding on low-rank (federated) training.
2. **Insufficient theoretical rationale:** The main point of the theoretical discuss points to the orthonormality of A as a way to have a stricter upper bound on the condition number of the problem. This analysis is insufficient in explaining the characteristics of the method for two reasons: (i) It holds on a very restrictive setting (logistic regression and one layer network) and (ii) does not reflect where the real advatage comes from experimentally. Indeed, the orhonormality of A does not appear as the main factor improving the performance (see Table 3). Better analysis should be provided to offer relevant insights on the induced dynamics of low-rank training
3. **Missing comparison with low-rank training works:** this paper compares with closely related lora-based methods, but does not discuss the relationship with related FL papers on low-rank training. For example [1] exploits low-rankness of the subspace spanned by gradients to drastically reduce the number of parameters transmitted through the network. That algorithm should be compliant with the premises of this work (avoid DP noise amplification or restricting expressiveness), and so it should be considered as competing baseline.


[1] Azam et. al, Recycling Model Updates in Federated Learning: Are Gradient Subspaces Low-Rank?, ICLR 2022

---

> ### Author Rebuttal · Authors · 2025-07-30
>
> We sincerely appreciate you for your time, effort, and recognizing the clear presentation and straightforward implementation, and we kindly acknowledge that other reviewers noted the **simple and effective design** (z79n, toPF, w3wU), **robust and consistent empirical results** (z79n, toPF, w3wU), **theoretical depth** (toPF), and **practical impact and timeliness** (toPF, w3wU). We address your concerns below and would be glad to dicuss additional points during discussion period:
>
> ---
> > **[Q1-1]** The motivation for recomputing SVD after training only B is not clear.
> - We respectfully note that the SVD step of our $\texttt{FedSVD}$  is **not an arbitrary choice**. It is the **mathematically optimal way** to factorize the aggregated knowledge from clients.
> - After the server aggregates the client updates to form $B_{i+1}$, the complete collective update is represented by the product $B_{i+1}\hat{A}_i$. $\texttt{FedSVD}$ uses SVD to refactorize this product.
> - As established by the **Eckart-Young-Mirsky theorem** [1], the truncated SVD provides the best possible rank-$r$ approximation of a matrix.
> - Since the rank of $B_{i+1}\hat{A}\_{i}$ is at most $r$, our SVD procedure is a **lossless operation** that perfectly preserves the aggregated update information while reparameterizing it into a new, synchronized pair of matrices, $\hat{A}\_{i+1}, \hat{B}\_{i+1}$ for the next round.
> - This is a **principled method for information preservation**, not a heuristic.
> - We will clarify the above in the revision.
> ---
> > **[Q1-2]** Clients could optimize A or B depending on the round.
> - We appreciate your insightful suggestion. Following your suggestion, we perform experiments where clients alternate between training either the matrix $A$ or $B$ in each round. We refer to this method as **FedRand**.
> - This approach has the same communication complexity as our proposed method and avoids the SVD computation. However, our empirical results in the following table show that this alternating training scheme leads to **a significant drop in performance** across most datasets.
>
>     **[Table R1. Comparison with FedRand.]**.
>     |Method|MNLI (matched)|MNLI (mismatched)|SST-2|QQP|QNLI|
>     |-|-|-|-|-|-|
>     |FedAvg|$65.44\pm6.14$|$67.02\pm5.93$|$89.41\pm1.35$|$58.58\pm5.26$|$60.69\pm6.04$|
>     |FFA|$55.56\pm8.58$|$56.39\pm8.93$|$91.41\pm0.86$|$64.35\pm 0.86$|$72.38\pm4.96$|
>     |FedRand|$42.14\pm7.20$|$41.73\pm7.02$|$90.68\pm0.59$|$55.15\pm5.57$|$49.46\pm0.00$|
>     |$\texttt{FedSVD}$|$\textbf{71.68}\pm3.18$|$\textbf{73.03}\pm2.89$|$91.31\pm0.85$|$\textbf{72.41}\pm2.36$|$\textbf{75.49}\pm4.20$|
> - Regarding the computational burden of SVD, we argue this cost is **manageable and can be made negligible**. First, as shown in `Fig. 5`, $\texttt{FedSVD}$ 's performance is robust to less frequent SVD computations. Second, we conduct new experiments during the rebuttal period using a randomized low-rank SVD approximation [2].
> - As shown below, this approach retains full accuracy while reducing the average **server-side runtime by over 98%** on the MNLI dataset.
>
>     **[Table R2. Experiments with low-rank SVD.]**
>     |Method|Matched Acc|Mismatched Acc|Avg. Run-time|
>     |-|-|-|-|
>     |$\texttt{FedSVD}$ w/ full SVD|$71.57\pm3.18$|$73.03\pm2.89$|$9.07\text{s}\pm0.07$|
>     |$\texttt{FedSVD}$ w/ low-rank SVD|$72.76\pm1.82$|$72.74\pm1.64$|$0.13\text{s}\pm0.06$|
> ---
> > **[Q2-1]** The theoretical analysis holds on a very restrictive setting.
> - Yes, this is a restricted setting, but at least provides **insight into why orthonormality can facilitate optimization**. This simplification of deep learning models into linear ones has been indeed explored in prior works [3, 4].
> - We believe that extending this theoretical analysis to more complex non-convex optimization settings remains an **important direction for future research**.
> ---
> >**[Q2-2]** Othonormality of A is not the main factor improving the performance (Table 3).
> - Thank you for pointing it out. In addition to the merit of orthonormality, we explain why periodically aligning the matrix $A$ with the principal direction of the aggregated update is crucial.
> - It ensures the model's updates are applied in the most effective and impactful way, preventing the learning process from being constrained by outdated or random directions.
> - We can formalize this core benefit. At the beginning of round $i+1$, the server has the aggregated client matrix, $B_{i+1}$ and the previous round's matrix $\hat{A}_{i}$.
> - The complete collective knowledge from the previous round is captured in the product $\Delta\_{i+1}=B\_{i+1}\hat{A}\_{i}$. The server's objective is to apply a rank-$r$ update, $\hat{B}\_{i+1}\hat{A}\_{i+1}$, that best reflects this collective knowledge. This is equivalent to finding the rank-$r$ matrix $\hat{B}\_{i+1}\hat{A}\_{i+1}$ that minimizes the approximation error with respect to $\Delta\_{i+1}$:
> $$
> \min\_{\hat{A}\_{i+1}, \hat{B}\_{i+1}}\lVert \Delta\_{i+1}- \hat{B}\_{i+1}\hat{A}\_{i+1} \rVert_F^2 \; \text{ subject to } \text{rank}(\hat{A}\_{i+1}) \leq r, \text{rank}(\hat{B}\_{i+1}) \leq r
> $$
> - The Eckart-Young-Mirsky [1] theorem provides the optimal solution to this problem. It states that the **best rank-$r$ approximation of a matrix $\Delta_{i+1}$ is obtained by its truncated SVD**. Our $\texttt{FedSVD}$  method implements exactly this solution.
> - Since $\texttt{FedSVD}$  defines the new matrices as $\hat{A}\_{i+1}=V^\top[:r, :]$ and $\hat{B}\_{i+1}=U[:, :r]\Sigma[:r, :r]$, where $U\Sigma V^\top = B\_{i+1}\hat{A}\_{i}$. Since the rank of $\Delta\_{i+1}$ is at most $r$ by the design of LoRA, the rank-$r$ SVD perfectly reconstructs the matrix, meaning $\Delta\_{i+1}=\hat{B}\_{i+1}\hat{A}\_{i+1}$.
> - Therefore, the SVD reparameterization is not just an arbitrary update rule. It is the **mathematically optimal strategy for preserving and propagating the aggregated knowledge** for all clients into the new low-rank adapters for the next round.
> ---
> >**[Q3]** A missing baseline [5] exploits low-rank of the subspace spanned by gradients.
> - We thank you for highlighting the highly relevant work by Azam et al [5]. While both our $\texttt{FedSVD}$ and their LBGM algorithm leverage the low-rank properties, we believe they **address orthogonal problems, making LBGM not a direct experimental baseline**. Specifically, our work focuses on **private federated learning in resource-constrained environments**, while LBGM is a general-purpose compression technique for full gradients.
> - We agree this is an important connection to make, and we will gladly add a **discussion of Azam et al. to `Section 2`** in a revision to better contextualize our contributions. The reasons why it is more suitable for discussion rather than direct experimental comparison are twofold:
> - $\texttt{FedSVD}$ is designed for the parameter efficient fine-tuning (PEFT) setting where fine-tuning large models makes the **computation or transmission of the full gradient infeasible** for clients. Our method operates exclusively on small LoRA adapter matrices. In contrast, LBGM's core mechanism requires clients to periodically **compute and transmit the entire gradient vector**, representing a fundamentally different operational paradigm. Comparing a PEFT method to a full-gradient compression technique would be an apples-to-oranges comparison.
> - Our primary contribution is to **mitigate the DP noise amplification** that is specific to the LoRA architecture in a private FL setting. The **LBGM framework is not designed for this context**.
> - Applying differential privacy to LBGM would necessitate **adding noise to the full, high-dimensional gradient vectors** during its periodic updates. This approach is known to severely degrade model performance [6], which is precisely the problem that private PEFT methods like $\texttt{FedSVD}$ are designed to overcome.
> - As noted, this will be incorporated into the revised `Section 2` for clarity.
> ---
> >**[Q4]** Other intuitive strategies for avoiding noise amplification.
> - Please refer to **[Q1-2]**.
> ---
> >**[Q5]** Better motivate why FedSVD does not limit the expressiveness of decomposition.
> - We appreciate this insightful question, which **highlights the core novelty of our approach**. While it's true that matrix $A$ is not trained directly with gradients, it is not fixed. Instead, it is adaptively reparameterized in each communication round. This dynamic updating process is precisely why our method avoids limiting the expressiveness of the low-rank decomposition, unlike methods that use a static matrix $A$.
> - Our method, $\texttt{FedSVD}$, updates $A$ on the server by performing a SVD on the aggregated update product from the previous round ($BA$). The new matrix $A$ is then re-initialized using the orthonormal $r$ right singular vectors ($V^{\top}$) obtained from this decomposition.
> - This means that $A$ is set to an orthonormal basis that captures the principal directions of the aggregated updates from all clients.
> ---
> >**[Q6-1]** Relation to priors works.
> - Please refer to **[Q3]**.
> ---
> >**[Q6-2]** Guarantee with low-rank training.
> - Please refer to **[Q2-2]**.
> ---
> ### References
> [1] Eckart, Carl, and Gale Young. "The approximation of one matrix by another of lower rank." Psychometrika. 1936.
>
> [2] Halko, Nathan, Per-Gunnar Martinsson, and Joel A. Tropp. "Finding structure with randomness: Probabilistic algorithms for constructing approximate matrix decompositions." SIAM. 2011.
>
> [3] Kawaguchi, Kenji. "Deep learning without poor local minima." NeurIPS. 2016.
>
> [4] Achour, El Mehdi, François Malgouyres, and Sébastien Gerchinovitz. "The loss landscape of deep linear neural networks: a second-order analysis." JMLR. 2024.
>
> [5] Azam, Sheikh Shams, et al. "Recycling model updates in federated learning: Are gradient subspaces low-rank?." ICLR. 2021.
>
> [6] Hölzl, Florian A., Daniel Rueckert, and Georgios Kaissis. "Equivariant differentially private deep learning: Why DP-SGD needs sparser models." ACM. 2023.

---

> > ### Comment · Reviewer_yjb1 · 2025-08-05
> >
> > I thank very much the authors for the detailed rebuttal, and especially for analyzing an alternative strategy to the one proposed in the paper.
> > However, there are still some points which are not convincing to me.
> >
> > **[Q1-1] Why recomputing SVD**
> > I acknowledge that the r-truncated SVD is the optimal way for r-rank approximation of a matrix, my doubt is on why we need to refactorize the BA product. I was referring to the motivation for this step when talking about heuristics, not to SVD itself.
> > To really represent a step forward, it should be clearly explained why the reparametrization is necessary, especially w.r.t to other simpler strategies (see below)
> >
> > **[Q1-2] Difference w.r.t FedRand**
> > I thank the author for providing these experiment with a simpler heuristic to avoid limiting the expressiveness of A and still avoid noise amplification. It is not clear to me why there is such a big gap of this FedRand variant and even vanilla FedAvg. I expected it to work similarly as doing LoRA on both matrices together (e.g. with appropriate number of local steps, the two solutions should not differ much). Can the author explain why there is such drop of performance? Are there other analyses to this regard?
> >
> > **[Q2-2] Orthonormality as main factor for performance**
> > This point is still not clear to me. In the theory, ortonormality of A is used to derive an advantage, but from experiments it is not the main factor. Could the authors clarify what is the precise reason for the improved performance and how this is capture by the presented theory?
> >
> > The other points have been well addressed.

---

> ### Author Response · Authors · 2025-08-06
>
> We sincerely appreciate your response to our rebuttal and your active participation during this discussion period, which has helped us better understand the core of your concerns. We hope the following response addresses your questions and resolves any remaining issues.
>
> ---
>
> > **[Q1]**  Why do we need to refactorize the product $BA$ ?
>
> - We sincerely appreciate you pointing this out. We clarify the motivation behind the refactorization as follows:
>
> - The motivation for refactorizing the product $BA$ using SVD is not merely a heuristic but a **principled approach to accelerate and stabilize** the optimization process in federated learning. Our method is designed to adaptively align the low-rank update with the most significant directions of the underlying data, which results in a better-conditioned optimization problem.
>
> - Our goal is to **minimize the upper bound** on the condition number of Hessian, which as shown in `Theorem 2.2` of our paper, is given by $$\frac{\lambda\_\text{max}(M\_k)}{\lambda\_\text{min}(M\_k|\_{\mathcal{R}(A^\top)})}.$$
>
> - To make this bound tighter, we must select the $r$-dimensional row space of $A$, denoted as $\mathcal{R}(A^\top)$, to maximize the smallest eigenvalue of the data covariance matrix $M\_k=\frac{1}{n\_k}\sum_{i=1}^{n\_k}\sigma(z\_i)(1-\sigma(z\_i))\mathbf{x}\_i\mathbf{x}^\top\_i$ when restricted to that subspace, i.e., maximize $\lambda\_\text{min}(M\_k|\_{\mathcal{R}(A^\top)})$.
>
> - According to the Courant-Fischer Theorem, the optimal $r$-dimensional subspace, $\mathcal{S}^*$, that maximizes this restricted eigenvalue is the one spanned by the eigenvectors corresponding to the $r$ largest eigenvalues of $M\_k$. For this optimal subspace, the minimum restricted eigenvalue is precisely $\lambda\_r(M\_k)$, the $r$-th largest eigenvalue of $M\_k$.
>
> - A simpler strategy might involve choosing $A$ as a fixed random matrix with orthonormal rows. For such a random subspace $S\_\text{rand}=\mathcal{R}(A^\top\_\text{rand})$, the expected minimum restricted eigenvalue is less than the optimal value, i.e., $\mathbb{E}[\lambda\_\text{min}(M\_k|\_{S\_\text{rand}})]<\lambda\_\text{min}(M\_k|\_{\mathcal{S}^*})=\lambda\_r(M_k)$ unless $\lambda\_1=\lambda\_2=\cdots=\lambda\_d$.
>
> - In contrast, $\texttt{FedSVD}$ adaptively chooses the subspace $\mathcal{S}\_\text{SVD}=\mathcal{R}(A^\top)$ by using the top-$r$ right singular vectors of the aggregated matrix $BA$. These vectors are also the top-$r$ eigenvectors of $(BA)^\top(BA)$. We argue that this is an excellent proxy for the optimal subspace $\mathcal{S}^*$.
>
> - We can formalize this relationship using the **Davis-Kahan Theorem** [1], which bounds the distance between the subspaces $\mathcal{S}^*$ and $\mathcal{S}_\text{SVD}$:
>
>     $$
>      \lVert\sin(\Theta(\mathcal{S}^*, \mathcal{S}\_\text{SVD})) \rVert\_F\leq \frac{\lVert M\_k - (BA)^\top BA\rVert_F}{\delta},
>     $$
>
> - where $\delta = \lambda\_r(M\_k) - \lambda\_{r+1}(M\_k)$ is the eigengap of $M_k$. We hypothesize that for an effective federated learning update, the aggregated update $BA$ must capture the structure of the underlying data, making the norm $\lVert M\_k - (BA)^\top (BA)\rVert\_F$ small as training goes on.
>
>
> - A small distance between $\mathcal{S}$ and $\mathcal{S}^\*$ implies that their geometric and algebraic properties are similar. By the  continuity of eigenvalues with respect to their underlying vector space, if $\mathcal{S}\_\text{SVD}$ is provably close to $\mathcal{S}^\*$, then their minimum restricted eigenvalues will also be close: $\lambda_\text{min}(M_k|\_{\mathcal{S}\_\text{SVD}}) \approx \lambda\_\text{min}(M\_k|\_{\mathcal{S}^*})=\lambda\_r(M\_k)$.
>
> - This leads to the key comparison: $\mathbb{E}[\lambda\_\text{min}(M\_k|\_{S\_\text{rand}})] < \lambda\_\text{min}(M\_k)\approx \lambda\_\text{min}(M\_k|\_{\mathcal{S}\_\text{SVD}})$.
>
> - From Jensen's inequality, $\mathbb{E}[f(X)]\geq f(\mathbb{E}[X])$, we have:
>
>     $$\mathbb{E}\left[\frac{1}{\lambda\_\text{min}(M\_k|\_{\mathcal{S}\_\text{rand}})}\right] \geq \frac{1}{\mathbb{E}[\lambda\_\text{min}(M\_k|\_{\mathcal{S}\_\text{rand}})]}.$$
>
> - Combining  these inequalities gives us a direct comparison of the inverse minimum restricted eigenvalues, which is the term that influences the condition number bound:
>
>     $$\frac{1}{\lambda\_\text{min}(M\_k|\_{\mathcal{S}\_\text{SVD}})}\approx\frac{1}{\lambda\_\text{min}(M\_k|\_{\mathcal{S}^*})} <   \frac{1}{\mathbb{E}[\lambda\_\text{min}(M\_k|\_{\mathcal{S}_\text{rand}})]} \leq \mathbb{E}\left[\frac{1}{\lambda\_\text{min}(M\_k|\_{\mathcal{S}\_\text{rand}})}\right].$$
>
> - This final inequality demonstrates that **the condition number bound for $\texttt{FedSVD}$ is tighter than the expected bound for a random approach**. By periodically refactorizing $BA$, we ensure that $A$ remains aligned with the principal components of the data throughout the training process, leading to a more stable and efficient optimization.
> ---

---

> ### Author Response · Authors · 2025-08-06
>
> ---
>
> > **[Q2]** It is not clear why there is such a big gap of this FedRand variant and even vanilla FedAvg.
>
> - We appreciate you pointing this out. We believe that the primary weakness of an alternating optimization strategy is that it relies on **stale information**.
> - Specifically, when the client matrix $B$ is optimized, it is done so using **a version of the global matrix $A$ that is fixed from the previous round**. This makes the optimization of $B$ inherently sub-optimal, as it is guided by incomplete information about the overall model's direction.
> - This problem is compounded by the random sampling of clients. Since there is no guarantee that **the same clients will be chosen in the next round**, the local optimization of $B$ for the current clients' data is performed using a matrix $A$ that was finalized based on data from a different set of clients in the previous round.
> - It might appear that these challenges also apply to $\texttt{FedSVD}$. However, our method is relatively free from these issues, **enabling faster convergence**.
> - Adaptive reparameterization of $\texttt{FedSVD}$ allows $A$ to capture the principal directions of aggregated updates, ensuring that local $B$ optimizations are guided by globally informed, up-to-date directional information rather than a noisy or outdated $A$.
>
> - We argue that this mismatch between **the current data and the stale matrix creates noisy and inefficient updates**, slowing the training process and lowering overall accuracy compared to our $\texttt{FedSVD}$.
>
> ---
>
> > **[Q3]** Orthonormality is not a main factor for performance improvement.
>
> - We thank the reviewer for this insightful comment. We agree that on the MNLI dataset, the performance gain from enforcing orthonormality is modest (nearly 1 percentage point), as shown in `Table3` of the main paper.
>
> - We acknowledge that a periodic SVD-based reparameterization, even without strict orthonormality, can offer some advantages by adaptively aligning $A$ with global update directions and improving flexibility over fixed A matrices. However, **achieving additional performance gains necessitates maintaining the orthonormal structure.**
>
> - As we proved in `Theorem 2.2`, the bound on the Hessian's condition number ensures more stable and accelerated convergence. Moreover, the orthonormality constraint bounds the gradient norm of $B$, as shown in `Equation (8)`, thereby preserving a more genuine update signal under DP-SGD.
>
> - In line with our theoretical findings, we have found that **the impact of orthonormality can be substantially more pronounced** depending on the dataset. To investigate this further, we conducted an additional experiment on the SNLI dataset. The results, presented below, show a significant performance improvement of **nearly 12 percentage points** when the orthonormal structure is maintained.
>
>
>     **[Table R3. Effect of orthonormality on SNLI.]**
>
>     |Method|Acc|
>     |-|-|
>     |$\texttt{FedSVD}_\text{w/o orthonormal}$|$69.48\pm9.45$|
>     |$\texttt{FedSVD}$|$\textbf{81.16}\pm2.37$
>
>
> - This significant improvement on SNLI provides **strong empirical support** for our theoretical analysis in `Theorem 2.2` that the orthonormal structure of matrix $A$ improves convergence. This demonstrates that orthonormality can be a critical factor for performance, validating the core design of our method.
>
> - We will include the above discussion in the revision.
>
> ---
>
> ### References
>
> [1] Davis, Chandler, and William Morton Kahan. "The rotation of eigenvectors by a perturbation. III." SIAM Journal on Numerical Analysis 7.1 (1970): 1-46.

---

> > ### Author Response · Authors · 2025-08-07
> >
> > Dear Reviewer yjb1
> >
> > As the discussion period is coming to a close, if there are any remaining concerns, we would be happy to clarify them further. Otherwise, if you feel that your concerns have been sufficiently addressed, we would be grateful if this could be reflected in your final score.
> >
> > Best Regards,
> >
> > The Authors

---

> > ### Comment · Reviewer_yjb1 · 2025-08-07
> >
> > I thank the authors for their insightful response, which addresses my core concerns. I would like to confirm that I have correctly understood the theoretical rationale behind the proposed approach.
> >
> > **[Q1]** Why do we need to refactor the product BA?\
> > In your message, you write that
> >
> > >“The optimal r-dimensional subspace $S^*$ that maximizes the minimum restricted eigenvalue is the one spanned by the eigenvectors corresponding to the top-r largest eigenvalues of $M_k$​,”
> >
> > and that FedSVD
> >
> > >is an excellent proxy for this optimal subspace” because the top-rr singular vectors of $BA$ are expected to be close to $S^*$.
> >
> > My understanding is that you are not claiming FedSVD is provably optimal, but rather that it is a practical approximation to the infeasible choice $S^*$, motivated by the expected alignment between the principal components of $M_k$ and the singular vectors of $BA$. Could you confirm if this interpretation is correct?
> >
> > If so, it might strengthen the work to empirically quantify the gap between FedSVD and $S^*$ in a controlled setup where $M_k$ can be computed (e.g. compare the eigenvalues, measure the distance between this optimal subspace, the one calculated by FedSVD and the r-random baseline).
> >
> > I believe such an ablation would substantially reinforce the justification for your method and clarify how closely FedSVD approximates the optimal direction in practice.
> >
> > I hope the authors will consider integrating this discussion into the revision and possibly including such an experiment. That said, your response already addresses my main concern, and I am happy to revise my score accordingly.
> >
> > **[Q2]** It is not clear why there is such a big gap of this FedRand variant and even vanilla FedAvg.\
> > Thanks for the explanations. I feel some more experiments on that would be beneficial, since the staleness depends on number of local steps and learning rate, so it should be controllable. Similarly, sampling can have an influence but, unless there is high heterogeneity between local datasets, I would expect to have a marginal effect. It would be nice to have a deeper analysis. However, the proposed points are all plausible explanations.
> >
> > **[Q3]** Orthonormality is not a main factor for performance improvement.\
> > Thanks.

---

> ### Author Response · Authors · 2025-08-08
>
> We sincerely appreciate your effort and constructive feedback. We hope the following responses address your remaining concerns.
>
> > **[Q4]** My understanding is that you are not claiming FedSVD is provably optimal, but rather that it is a practical approximation to the infeasible choice
> $\mathcal{S}^*$, motivated by the expected alignment between the principal components of $M_k$ and the singular vectors of $BA$. Could you confirm if this interpretation is correct?
>
> - Thank you for clarification and your understanding our position. Yes, your interpretation is entirely correct. Our claim is that $\texttt{FedSVD}$ serves as a practical yet principled approximation to the optimal subspace $\mathcal{S}^*$.
>
> ---
>
> > **[Q5]** If so, it might strengthen the work to empirically quantify the gap between FedSVD and $\mathcal{S}^*$ in a controlled setup where $M_k$ can be computed.
>
> - We sincerely appreciate your thoughtful consideration and constructive suggestions to improve our paper. While our theoretical analysis uses distance between the subspace $\mathcal{S}^*$ and $\mathcal{S}_\text{SVD}$, a simple logistic regression setup allows to directly compute the **actual condition number** during optimization. We believe this is a more direct and practical measure of our method's effectiveness.
>
> - Therefore, we directly measure the condition number $\kappa\_2 (H(B;A)):=\lambda_\text{max}(H(B;A))/\lambda_\text{min}(H(B;A))$ of $\texttt{FedSVD}$ and that of FFA-LoRA.
> - Specifically, we (1) extract features $X \in \mathbb{R}^{n \times d_\text{in}}$ using a pretrained BERT model, (2) train a logistic regression head on top of these frozen features, and (3) then directly compute the Hessian's condition number, during optimization.
>
> - As shown in the table below, we observe that the condition number of $\texttt{FedSVD}$ is **consistently smaller** than that of FFA-LoRA throughout training, confirming its practical benefit for optimization.
>
>     **[Table R4. Comparison on condition numbers.]**
>     |Method|0|1k|2k|3k|4k|5k|
>     |-|-|-|-|-|-|-|
>     |FFA-LoRA|10.18|10.15|9.78|10.09|10.13|10.23|
>     |$\texttt{FedSVD}$|**1.67**|**1.52**|**1.51**|**1.50**|**1.50**|**1.51**|
>
> - Once again, we **sincerely appreciate your valuable feedback**. We will **incorporate all of this experiment and discussion** into the revision to strengthen the clarity and rigor of our paper.
>
> ---
>
>
> > **[Q6]** Sampling can have an influence but, unless there is high heterogeneity between local datasets, I would expect to have a marginal effect.
> - We fully agree that sampling effects are influenced by the level of heterogeneity across local datasets. We kindly note that our **experimental setup already accounts for particularly pronounced (high) heterogeneity**.
> - For example (`Table R8`), in the MNLI dataset with $\alpha = 0.5$, **Client 2** is heavily biased towards Label 0, whereas **Client 1** has only 2.5% of Label 0. Conversely, for Label 1, **Client 3** holds 40.5%, while **Client 2** has just 0.7%.
>
>     **[Table R5. Per-label data distribution across clients (%) in the MNLI dataset ($\alpha=0.5$).]**
>     |Label|Client 0|Client 1|Client 2|Client 3|Client 4|Client 5|
>     |-|-|-|-|-|-|-|
>     |$0$|0.104|0.025|**0.673**|0.024|0.1|0.073|
>     |$1$|0.049|0.016|0.007|0.405|0.058|**0.465**|
>     |$2$|**0.333**|0.168|0.113|0.0001|0.064|0.322|
>
> - This pattern of **severe label imbalance** is consistently observed across all clients, and we expect such disparities to become even more pronounced as the number of clients increases.
> - We will clarify the per-label data distribution across clients for each dataset in the revision.

---

> ### Author Response · Authors · 2025-08-09
>
> > **[Q7]** I feel some more experiments on that would be beneficial, since the staleness depends on number of local steps and learning rate, so it should be controllable.
>
> - Thank you for this insightful suggestion. While it is intuitive that staleness could be controlled by tuning hyperparameters like the number of local steps, our experiments show this **relationship isn't straightforward** for the FedRand baseline. We respectfully argue that the negative effects of staleness in this alternating optimization scheme are not easily mitigated by simply adjusting this parameter.
>
> - To investigate, we evaluated FedRand with different numbers of local steps ($\tau \in \\{5, 10, 20\\}$), where $\tau = 10$ is the default in our main paper.
>
> - As shown in `Table R6`, performance is **non-monotonic**. The default setting of $\tau=10$ is significantly better than both fewer ($\tau=5$) and more ($\tau=20$) local steps on most tasks. This suggests a complex trade-off where increasing local steps may exacerbate the issue of optimizing with a stale global matrix, rather than resolving it.
>
>     **[Table R6. FedRand with varying number of local updates ($\tau$).]**
>     |local steps ($\tau$)|MNLI (Matched)|MNLI (Mismatched)|SST-2|QQP|QNLI|
>     |-|-|-|-|-|-|
>     |$5$|$33.09\pm0.85$|$33.18\pm0.76$|$50.18\pm0.62$|$48.57\pm8.40$|$49.89\pm0.36$|
>     |$10$|$42.14\pm7.20$|$41.73\pm7.02$|$90.68\pm0.59$|$55.15\pm5.57$|$49.46\pm0.00$|
>     |$20$|$34.17\pm1.09$|$34.08\pm0.99$|$49.82\pm0.62$|$52.72\pm8.88$|$50.12\pm 0.38$|
>
> - We will include the FedRand results in the revised version of our paper.

---

> > ### Comment · Reviewer_yjb1 · 2025-08-09
> > **Thank you for additional controlled experiments**
> >
> > I would like to thank the authors for having constructively participated to the discussion. The provided experiments strenghen the motivation of the work and provide an empirical verification of the theoretical claims. Having some experiments comparing with an "ideal" solution would be a plus: the provided experiments are direct validation of the main theoretical claim, but do not
> > quantify the distance from the upper bound. If possible, I would advise the authors of thinking to this kind of experiments, because they will clarify how much margin of improvement is still there in practice, and could serve as guide for future work.
> >
> > The experiments on the FedRand baseline are interesting, I would advise to also take into account the learning rate for the revised version of the paper, as it may play a role. However, it is sufficiently clear there is a significant performance gap that is very unlikely to be filled by extensive tuning.
> >
> > I have revised my score and final recomendation towards acceptance.
> > Best.

---

> ### Author Response · Authors · 2025-08-09
>
> - We sincerely thank you for your constructive participation in the discussion and for your encouraging final recommendation.
>
> - In response to your suggestions, we are currently conducting the following experiments:
>
> 1. Comparison of the condition number between the optimal solution and $\texttt{FedVD}$
> 2. Evaluating the effect of learning rate variation for the FedRand baseline on MNLI with learning rates 0.2 and 1.0.
> As the discussion period ends in less than two hours, we will do our best to complete these runs and share preliminary results if possible. Should we be unable to finish in time, we will include the full results in the revised version of the paper.
>
> Thank you once again for your valuable feedback and support.
>
> Best regards,
>
> The Authors

---

> ### Author Response · Authors · 2025-08-09
>
> > **[Q8]** Experiments comparing with an "ideal" solution would be a plus.
>
> - We sincerely appreciate your time and patience, and report the results of 1) the **comparison of condition number, $\kappa\_2 (H(B;A)):=\lambda_\text{max}(H(B;A))/\lambda_\text{min}(H(B;A))$, between oracle and $\texttt{FedSVD}$**, and 2) **the effect of learning rate** for the FedRand baseline on MNLI with learning rates 0.2 and 1.0, as shown below:
>
>     **[Table R7. Comparison on condition numbers.]**
>     |Method|0|1k|2k|3k|4k|5k|
>     |-|-|-|-|-|-|-|
>     |FFA-LoRA|10.18|10.15|9.78|10.09|10.13|10.23|
>     |$\texttt{FedSVD}$|**1.67**|**1.52**|**1.51**|**1.50**|**1.50**|**1.51**|
>     |oracle|$1.06$|$1.01$|$1.02$|$1.04$|$1.03$|$1.02$|
>
>
>
> - In `Table R7`, we observe that the condition numbers of $\texttt{FedSVD}$ are already comparable to those of the oracle. However, we acknowlege that these results are obtained in a limited setting. Therefore, we believe there is future work to further improve $\texttt{FedSVD}$ in more realistic settings.
>
> ---
> > **[Q9]** Take into account the learning rate for FedRand.
>
> - Due to the time limit, we conducted the experiments on MNLI dataset with only a single seed in Table `R8`. We observe that the random baseline, FedRand, is also not easily improved through learning rate tuning.
>
>     **[Table R8. FedRand with varying learning rates.]**
>
>     |learning rate|MNLI (Matched)|MNLI (Mismatched)|
>     |-|-|-|
>     |$0.2$|$32.73$|$32.95$|
>     |$0.5$|$42.14\pm7.20$|$41.73\pm7.02$|
>     |$1.0$|$32.73$|$32.95$|
>
>
> - As you suggested, we believe these results empirically support our claims throughout the paper and further strengthen our work.
> - Once again, we sincerely appreciate your constructive suggestions, and we will complete these experiments and include them in the revision.

---

### Official Review · Reviewer_toPF · 2025-07-02

**Clarity:** 4
**Significance:** 2
**Originality:** 2
**Rating:** 4
**Confidence:** 3

**Summary:**

The paper introduces FedSVD, a method designed to improve the efficiency and stability of federated learning (FL) with Low-Rank Adaptation (LoRA) under differential privacy (DP). The key innovation is a global reparameterization based on singular value decomposition (SVD), which allows the matrix $A$ to adapt over time while mitigating noise amplification. In this approach, clients optimize and transmit only the $B$ matrix, while the server aggregates these updates, computes $BA$, and then refactors it via SVD to produce a new adaptive $\hat{A}$ and updated $B$ based on the clients' data. The paper claims that FedSVD consistently improves performance across various privacy settings and benchmarks, outperforming relevant baselines in both private and non-private regimes. Empirical results on GLUE tasks demonstrate the effectiveness of FedSVD.

**Questions:**

- How does FedSVD fundamentally differ from the decomposition-based federated learning methods in [1, 2,3 ] (or other similar studies)?
  - Can you clarify the unique contributions of FedSVD beyond its application to LoRA?
- How does the privacy-utility trade-off of FedSVD compare with other recent DP-FL methods? Could you provide a more comprehensive evaluation of privacy budgets across different methods?
- Does the additional communication cost of broadcasting $\hat{A}$ after each SVD step become a bottleneck in large-scale settings? How does this compare to existing lightweight methods like FFA-LoRA? (Could this be beneficial for federated fine-tuning a.k.a. maml or partial learning?)
- Could you evaluated FedSVD on larger or more complex datasets beyond GLUE? (e.g., SNLI[4], HellaSwag[5], MATH[6] )

[4] A large annotated corpus for learning natural language inference (2015)

[5] Hellaswag: Can a machine really finish your sentence? (2019)

[6] Measuring mathematical problem solving with the math dataset (2021)

**Ethical Concerns:**

["NO or VERY MINOR ethics concerns only"]

**Final Justification:**

Based on the author's additional explanation of the novelty and the extended experimental efforts, I have decided to raise my overall score.

**Limitations:**

The authors have discussed some limitations of the work in the conclusion.

**Quality:**

3

**Strengths And Weaknesses:**

### Strengths:
1. The introduction of SVD-based reparameterization to mitigate noise amplification appears to be an effective approach in DP settings.
2. The paper provides deep theoretical analysis showing how orthonormal rows of $A$ can improve the condition number of the Hessian, leading to better optimization dynamics.
3. Empirical evaluations on GLUE tasks demonstrate consistent performance improvements over baselines.

### Weaknesses:
1. A major concern is the novelty. There have been other studies on decomposition-based FL approaches in DP settings [1, 2, 3] that are not compared in the paper. The novelty here is mostly limited to applying these techniques to the new problem domain of LoRA training.
2. Although the authors claim that FedSVD improves performance across various DP settings, the privacy budget comparison with recent methods is not provided.
3. While the method introduces additional communication costs of broadcasting newly initialized $\hat{A}$ matrix to clients after each SVD step, the performance improvement seems to be marginal compared to the prior method FFA-LoRA.
4. While the paper demonstrates effectiveness on GLUE tasks, it remains unclear how well FedSVD scales to larger datasets or more complex models.

[1] Privacy-Preserving Distributed SVD via Federated Power (FedPower), CoRR 2021

[2] Privacy-Preserving Federated Singular Value Decomposition, Applied Sciences 2023

[3] Exploring Gradient Subspaces: Addressing and Overcoming LoRA's Limitations in Federated Fine-Tuning of Large Language Models, AAAI 2025

---

> ### Author Rebuttal · Authors · 2025-07-31
>
> We sincerely appreciate you for recognizing the theoretical soundness and performance consistency of our approach, and we kindly acknowledge that other reviewers emphasize **effective and simple design** (z79n, yjb1, w3wU), **robust experiments** (z79n, w3wU), **clear and well-written presentation** (z79n, yjb1, w3wU), and **practical relevance and timeliness** (yjb1, w3wU). We address your concerns below and would be glad to dicuss additional points during discussion period.
>
> ---
> >**[Q1]** A major concern is the novelty. There have been other studies on decomposition-based FL approaches in DP settings [1, 2, 3] that are not compared in the paper. The novelty here is mostly limited to applying these techniques to the new problem domain of LoRA training.
> - Thank you for your constructive feedback and providing relevant literature. However, we **respectfully disagree with the assessment that our novelty is limited** with respect to [1, 2, 3].
> - The cited works address **fundamentally different problems** from our own. The fundamental difference between our approach and FedPower, is that FedSVD is a model adaptation algorithm, whereas FedPower is a data analysis algorithm. We believe our contribution is a novel algorithmic solution to a specific challenge in FL with DP, not merely an application of existing techniques to LoRA.
> - The core novelty of our $\texttt{FedSVD}$ is its **use of SVD as a server side, postprocessing step to reparameterize LoRA adapters** $A$ and $B$ to solve the problem of optimization dynamics due to using fixed $A$ and noise amplicafication that arises when combining LoRA and DP-SGD.
> - Specifically, Guo et al. [1] and Liu et al. [2] focus on computing the **SVD of a large data matrix** distributed across clients with privacy guarantees, which is **not the same as optimizing neural network parameters**.
> - In contrast, $\texttt{FedSVD}$ performs a standard, centralized SVD on the small, aggregated LoRA adapters $BA$ on the server. Rather than analyzing client data, we use SVD to **re-initialize one of the LoRA adapters**, $A$, to align with the principal directions of the aggregated updates.
> - This process is designed to **avoid noise amplification** and, as our analysis shows, simultaneously helps to **stabilize training and accelerate convergence** with formal DP guarantee. These are fundamentally different goals and technical challenges.
> - Mahla et al. [3] fine-tunes all up-projection layers with GaLore [4], which significantly increases training and communication cost due to the vastly larger number of trainable parameters (e.g., **from 393,216 in our LoRA setup to 357,199,876**). This is **infeasible in our problem setting**, because individual clients often lack the computational and memory capacity required for full fine‑tuning of large models, as noted in `L39–40`.
> - Moreover, Mahla et al. [3] did not target DP guarantees, and a **larger number of parameters can make DP‑SGD unstable** [5].
> - On the other hand, FedSVD is designed for the highly parameter-efficient LoRA framework, making it suitable for resource-constrained federated learning environments, while providing a formal privacy guarantee.
> ### Summary
> - Guo et al. [1], Liu et al. [2], and Mahla et al. [3] are relevant to our work; however, they address **fundamentally different problems** than ours and are **not applicable to our problem setting**.
> - The fundamental difference between our approach and FedPower, is that FedSVD is a model adaptation algorithm, whereas FedPower is a data analysis algorithm.
> - We will clarify this difference in the revision.
> ---
> >**[Q2]** Although the authors claim that FedSVD improves performance across various DP settings, the privacy budget comparison with recent methods is not provided.
> - Thank you for pointing this out. We conducted an additional experiment on the GLUE benchmark using $\epsilon=3$, $\delta=10^{-5}$, with the same experimental setup used in `Table 2`:
>
>     **[Table R1. Experiments with $\epsilon=3,\delta=10^{-5}$.]**
>     |Method|MNLI (matched)|MNLI (unmatched)|SST-2|QQP|QNLI|Avg|
>     |-|-|-|-|-|-|-|
>     |FedAvg|$50.45\pm14.35$|$50.82\pm14.59$|$50.64\pm0.55$|$57.91\pm7.31$|$50.11\pm0.37$|$51.99\pm5.11$|
>     |FFA|$57.39\pm9.32$|$59.00\pm9.42$|$\textbf{90.69}\pm{1.23}$|$70.88\pm2.23$|$74.59\pm1.70$|$70.51\pm3.30$|
>     |$\texttt{FedSVD}$ (ours)|$\textbf{69.84}\pm5.36$|$\textbf{70.94}\pm5.40$|$89.93\pm0.80$|$\textbf{73.67}\pm1.67$|$\textbf{77.25}\pm2.09$|$\textbf{76.32}\pm2.29$|
>
>
> - We observe that **FedAvg collapses to near random guess** under the stricter privacy constraint, demonstrating its sensitivity to reduced $\epsilon$.
> - In contrast, $\texttt{FedSVD}$ **continues to outperform all baselines** even at $\epsilon=3$, highlighting its robustness to stronger privacy constraints.
> - We will include this result and discussion in the revision.
> ---
> >**[Q3]** While the method introduces additional communication costs of broadcasting newly initialized matrix to clients after each SVD step, the performance improvement seems to be marginal compared to the prior method FFA-LoRA.
> - Thank you for your comments on efficiency and performance. We would like to respectfully clarify what we believe is a misunderstanding on two points: **1) communication cost** and **2) significance of the performance improvement**.
> - Our proposed method, $\texttt{FedSVD}$, is designed to have the exact **same communication cost as the FFA-LoRA**. While we acknowledge in `L320-329` of `limitations` that broadcasting the newly initialized $A$ matrix is a possibility, we also describe the efficient mechanism used in our experiments to avoid this cost.
> - In this approach, the server **transmits only the aggregated $B$ matrix**, and each client locally reconstructs the new $A$ matrix, thereby incurring no additional communication overhead compared to FFA-LoRA.
> - While $\texttt{FedSVD}$ introduces additional overhead due to the SVD step, as discussed in `Section 4`, **this cost can be substantially mitigated** by leveraging randomized low-rank approximation techniques, such as the method described in [6].
> - As demonstrated in the table below, this approximation maintains performance while **significantly reducing the runtime of the SVD operation**. Moreover, as shown in `Fig. 5`, the computational cost can be further reduced by performing the SVD step less frequently.
>
>     **[Table R2.Experiemtns with low-rank SVD.]**
>     |Method|Matched Acc|Mismatched Acc|Avg. Run-time|
>     |-|-|-|-|
>     |$\texttt{FedSVD}$ w/ full SVD|$71.57\pm3.18$|$73.03\pm2.89$|$9.07\text{s}\pm0.07$|
>     |$\texttt{FedSVD}$ w/ low-rank SVD|$72.76\pm1.82$|$72.74\pm1.64$|$0.13\text{s}\pm0.06$|
> - Under DP constraints ($\epsilon=6,\delta=10^{−5}$) in `Table 2`, $\texttt{FedSVD}$ achieves an average accuracy of 76.79%, which is a substantial **8.77% improvement** over FFA-LoRA's 68.02%.
> - Furthermore, in additional experiments (see our answer to **[Q2]**) under an even stricter privacy budget of $\epsilon=3$, $\texttt{FedSVD}$ continues to significantly **outperform FFA-LoRA by 5.81%** in average accuracy (76.32% vs. 70.51%).
> ---
> >**[Q4]** While the paper demonstrates effectiveness on GLUE tasks, it remains unclear how well FedSVD scales to larger datasets (SNLI[7], HellaSwag[8], MATH[9]) or more complex models.
> - Following your suggestion, we first fine-tune RoBERTa-large on SNLI dataset [7] under DP constraints $(\epsilon=6, \delta=10^{-5})$ and report accuracy on the table below. FedSVD  again outperforms the baselines on the SNLI dataset.
>
>     **[Table R3. SNLI with RoBERTa-large.]**
>     |Method|Acc|
>     |-|-|
>     |FedAvg|$74.00\pm1.82$|
>     |FFA|$73.98\pm4.23$|
>     |$\texttt{FedSVD}$|$\textbf{81.16}\pm2.37$|
> - Furthermore, we fine-tune a SmolLM-360M model [10] on more complex dataset, HellaSwag [8] under DP constraints ($\epsilon=6, \delta=10^{-5}$).
>
>
> - FedSVD again outperforms other baselines under DP-constraints with $\epsilon=6$ and $\delta=10^{-5}$. The results highlight the core strengths of our method. FedAvg's performance degrades significantly due to the noise amplification from the DP-SGD. FFA-LoRA improves upon FedAvg by fixing one of LoRA adapters $A$. FedSVD provides further improvement by adaptively aligning $A$ with principal directions of aggregated updates, while also avoiding noise amplification.
>
>     **[Table R4. HellaSwag with SmolLM-360M.]**
>     |Method|Acc|
>     |-|-|
>     |FedAvg|$48.81\pm 0.28$|
>     |FFA-LoRA|$49.76\pm0.09$|
>     |$\texttt{FedSVD}$|$51.10\pm0.16$|
> ---
>
> ### References
>
>
> [1] Guo, Xiao, et al. "Privacy-preserving distributed SVD via federated power." arXiv. 2021.
>
> [2] Liu, Bowen, Balázs Pejó, and Qiang Tang. "Privacy-preserving federated singular value decomposition." Applied Sciences. 2023.
>
> [3] Mahla, Navyansh, Kshitij Sharad Jadhav, and Ganesh Ramakrishnan. "Exploring Gradient Subspaces: Addressing and Overcoming LoRA's Limitations in Federated Fine-Tuning of Large Language Models." AAAI. 2025.
>
> [4] Zhao, Jiawei, et al. "GaLore: Memory-Efficient LLM Training by Gradient Low-Rank Projection." ICML. 2024.
>
> [5] Hölzl, Florian A., Daniel Rueckert, and Georgios Kaissis. "Equivariant differentially private deep learning: Why DP-SGD needs sparser models." ACM Workshop. 2023.
>
> [6] Halko, Nathan, Per-Gunnar Martinsson, and Joel A. Tropp. "Finding structure with randomness: Probabilistic algorithms for constructing approximate matrix decompositions." SIAM. 2011.
>
> [7] Bowman, Samuel R., et al. "A large annotated corpus for learning natural language inference." ACL. 2015.
>
> [8] Zellers, Rowan, et al. "Hellaswag: Can a machine really finish your sentence?." ACL. 2019.
>
> [9] Hendrycks, Dan, et al. "Measuring mathematical problem solving with the math dataset." NeurIPS. 2021.
>
> [10] Allal, Loubna Ben, et al. "SmolLM2: When Smol Goes Big--Data-Centric Training of a Small Language Model." arXiv. 2025.

---

> ### Comment · Reviewer_toPF · 2025-08-03
> **Thank you for the detailed answer**
>
> The author's response effectively addresses the main concerns I initially had: (1) it strengthens the explanation of the novelty of the work, (2) it includes additional experiments on differential privacy (DP) budget, and (3) it provides further empirical evidence to support the method’s scalability. Based on these improvements, I have decided to raise my overall score. That said, I believe that the discussion on the novelty and contribution of the paper could be further clarified and emphasized in the main text, which would help readers better appreciate the significance of the work.

---

> ### Author Response · Authors · 2025-08-06
>
> We sincerely appreciate your positive decision and encouraging comments on our paper. We assure you that we will clarify and emphasize the novelty and contributions in relation to our response to **[Q1]**, and we will also include the additional experiments in the revision.

---

### Official Review · Reviewer_z79n · 2025-07-03

**Clarity:** 4
**Significance:** 3
**Originality:** 3
**Rating:** 5
**Confidence:** 3

**Summary:**

This paper proposes a new FL algorithm for LoRA training, FedSVD. FedSVD uses reparameterization of LoRA adopter matrices BA based on SVD and only requires clients to train a matrix B, reducing the amount of noise under DP while keeping the learning capability. Through empirical experiments, the authors show that their proposed algorithm outperforms the baseline LoRA algorithms.

**Questions:**

- What are the baseline results for non-finetuned models and centralized finetuned models?
- The performance of the proposed method when K=12 in Figure 4(b) is significantly better than the baseline. Do the authors have any intuition behind the results? Do they think the gap becomes more significant when we have larger number of clients?
- Do the authors try different DP parameters from $\epsilon=6$? Why not sweeping $\epsilon$ values?

**Ethical Concerns:**

["NO or VERY MINOR ethics concerns only"]

**Final Justification:**

The authors addressed most of the concerns/questions that I raised in my review, except for the limited model variation. While I still believe the experiment on larger models is insightful and natural thing to do, the rest of contributions made in the paper outweigh the weakness. As such, I would keep my recommendation for accepting the paper.

**Limitations:**

yes

**Quality:**

4

**Strengths And Weaknesses:**

Strengths
- The paper is clearly written
- The idea of using SVD in LoRA training is simple but effective—the experiment results support the superiority
- The experiment demonstrates the validity of the algorithm through multiple angles. In particular, the experiments listed in Section 3.3 support that the proposed algorithm is robust and its components, e.g., orthonormal initialization of A, are effective respectively.

Weaknesses
- The experiment uses only one model, RoBERTa-large, which is much smaller than the recent models such as Llama. Running experiments on larger models would validate that the scalability of the proposed method.

---

> ### Author Rebuttal · Authors · 2025-07-31
>
> We sincerely appreciate you for your time, effort, and recognizing the simple-yet-effective design and robust experimental validation, and we kindly acknowledge that other reviewers emphasize **theoretical soundness** (toPF), **clear and well-written presentation** (w3wU, yjb1), **straightforward implementation** (yjb1, w3wU), **performance consistency** (toPF), and **practical relevance and timeliness** (toPF, yjb1, w3wU). We address your concerns below and would be glad to dicuss additional points during discussion period.
>
> ---
> >**[Q1]** The experiment uses only one model, RoBERTa-large, which is much smaller than the recent models such as Llama. Running experiments on larger models would validate that the scalability of the proposed method.
> - We agree that experiments with larger models would help validate the scalability of $\texttt{FedSVD}$. However, we assume that **individual clients operate under limited computational and memory resources**.
> - This **low-resource assumption** is commonly adopted in the federated learning literature, for example, in mobile-device federated learning scenarios [1, 2, 3, 4].
> - In such realistic settings, deploying billion-scale language models like LLaMA is usually not feasible.
> - Instead, we extend our experiments to more complex data. Specifically, we use the HellaSwag benchmark [5] along with SmolLM-360M [6], a recent and efficient model. This setup demonstrates the scalability of $\texttt{FedSVD}$ with respect to data size and task complexity.
> - As shown in the table below, FedSVD again outperforms other baselines under DP-constraints with $\epsilon=6$ and $\delta=10^{-5}$. The results highlight the core strengths of our method. FedAvg's performance degrades significantly due to the noise amplification from the DP-SGD. FFA-LoRA improves upon FedAvg by fixing one of LoRA adapters $A$. FedSVD provides further improvement by adaptively aligning $A$ with principal directions of aggregated updates, while also avoiding noise amplification.
>
>     **[Table R1. HellaSwag with SmolLM-360M]**
>     |Method|Acc|
>     |-|-|
>     |FedAvg|$48.81\pm 0.28$|
>     |FFA-LoRA|$49.76\pm0.09$|
>     |$\texttt{FedSVD}$|$\textbf{51.10}\pm0.16$|
> ---
> >**[Q2]** What are the baseline results for non‑finetuned models and centralized finetuned models?
> - Thank you for the question. GLUE consists of text classification tasks, so a model **without task‑specific fine‑tuning** (e.g., a randomly initialized linear head) performs around **random/majority‑class** levels: roughly **$\approx33\%$** for 3‑way tasks (e.g., MNLI) and **$\approx50\%$** for balanced binary tasks (e.g., SST‑2). For imbalanced binaries (e.g., QQP), the majority‑class baseline can be slightly higher.
> - We report the performance of a **centralized full fine‑tuned** model by training RoBERTa‑Large on GLUE without FL or DP and then evaluating as follows:
>
>     **[Table R2. Centralized full fine-tuned model.]**
>     |MNLI (matched)|MNLI (unmatched)|SST-2|QQP|QNLI|Avg|
>     |-|-|-|-|-|-|
>     |$90.2$|$90.2$|$96.4$|$92.2$|$94.7$|$92.74$|
> - We will include the above as the **upper bound** in the revision.
> ---
> >**[Q3]** The performance of the proposed method when $K=12$ in Figure 4(b) is significantly better than the baseline. Do the authors have any intuition behind the results?
> - Thank you for the insightful question. Our core intuition is that the performance gap widens with a larger client pool because FedSVD's adaptive mechanism creates a shared, well-conditioned projection matrix $A$ that is derived from the collective updates of the diverse clients using SVD. This provides a stable optimization landscape for all participants, a task where static baselines increasingly underperforms as data heterogeneity grows.
> - FedSVD leverages diversity to create a data-driven **consensus projection**. With a larger client pool, the server aggregates $B$ matrices from a more heterogeneous set of local data distributions. It then performs SVD on the product of the aggregated $B$ and the previous $A$. The new shared $A$ is composed of the orthonormal right singular vectors from this operation, allowing it to **better capture the principal directions of the aggregate updates** across the entire federation.
> - This data-driven consensus projection is crucial for heterogeneous data. Our theoretical analysis shows that using an orthogonal $A$ improves the condition number of the Hessian matrix for the local optimization of B. This creates a **better-behaved optimization landscape** that promotes **faster and more stable convergence** for every client.
> - In contrast, a baseline like FFA-LoRA uses a fixed, randomly initialized matrix for A, which becomes an increasingly poor, one-size-fits-none projection as client diversity grows, thus hindering performance.
> ---
> >**[Q4]** Do they think the gap becomes more significant when we have larger number of clients?
> - Yes, we hypothesize that **the performance gap will become more significant with a larger number of clients**. As the client pool and data heterogeneity grow, the limitation of a non-adaptive projection matrix becomes a more severe bottleneck.
> - Conversely, FedSVD's adaptive SVD mechanism receives richer information from the diverse updates, enabling it to construct an even more robust and generalized A matrix. This should further widen the performance gap between our method and the baselines.
> ---
> >**[Q5]** Try different DP parameters, other than $\epsilon=6$.
> - Thank you for pointing this out. We conducted an additional experiment on the GLUE benchmark using $\epsilon=3$, $\delta=10^{-5}$, with the same experimental setup used in `Table 2`:
>
>     **[R3. GLUE experiments with $\epsilon=3,\delta=10^{-5}$.]**
>     |Method|MNLI (matched)|MNLI (unmatched)|SST-2|QQP|QNLI|Avg|
>     |-|-|-|-|-|-|-|
>     |FedAvg|$50.45\pm14.35$|$50.82\pm14.59$|$50.64\pm0.55$|$57.91\pm7.31$|$50.11\pm0.37$|$51.99\pm5.11$|
>     |FFA|$57.39\pm9.32$|$59.00\pm9.42$|$\textbf{90.69}\pm{1.23}$|$70.88\pm2.23$|$74.59\pm1.70$|$70.51\pm3.30$|
>     |$\texttt{FedSVD}$ (ours)|$\textbf{69.84}\pm5.36$|$\textbf{70.94}\pm5.40$|$89.93\pm0.80$|$\textbf{73.67}\pm1.67$|$\textbf{77.25}\pm2.09$|$\textbf{76.32}\pm2.29$|
>
> - We observe that FedAvg collapses to near random guess under the stricter privacy constraint, demonstrating its sensitivity to reduced $\epsilon$.
> - In contrast, $\texttt{FedSVD}$ continues to outperform all baselines even at $\epsilon=3$, highlighting its robustness to stronger privacy constraints.
> - We will include this result and discussion in the revision.
> ---
> ### References
>
> [1] Kairouz, Peter, et al. “Advances and Open Problems in Federated Learning.” Foundations and Trends in Machine Learning. 2021.
>
> [2] Bonawitz, Keith, et al. “Towards Federated Learning at Scale: System Design.” SysML. 2019.
>
> [3] Hard, Andrew, et al. “Federated Learning for Mobile Keyboard Prediction.” arXiv. 2018.
>
> [4] McMahan, H. Brendan, et al. “Communication‑Efficient Learning of Deep Networks from Decentralized Data.” AISTATS. 2017.
>
> [5] Zellers, Rowan, et al. "HellaSwag: Can a machine really finish your sentence?." ACL. 2019.
>
> [6] Allal, Loubna Ben, et al. "SmolLM2: When Smol Goes Big--Data-Centric Training of a Small Language Model." arXiv 2025.

---

> > ### Comment · Reviewer_z79n · 2025-08-04
> >
> > I thank the authors for addressing the concerns I raised in the review. I believe some additional points and experimental results in the authors' response further improve the paper quality and clarity. While I support for the acceptance of this paper as it is, the discussions (and supporting experiments) for Q3&4 would be interesting enough to add to the paper in the revision since they should be the scenarios where the proposed method shines more.

---

> ### Author Response · Authors · 2025-08-06
>
> We sincerely appreciate **your efforts and participation during this discussion period**. We respond to your additional suggestion below:
>
> ---
>
> > I thank the authors for addressing the concerns I raised in the review. I believe some additional points and experimental results in the authors' response further improve the paper quality and clarity. While I support the acceptance of this paper as it is, the discussions (and supporting experiments) for Q3&4 would be interesting enough to add to the paper in the revision since they should be the scenarios where the proposed method shines more.
>
>
> - We are sincerely grateful for your positive evaluation and your encouraging comments. We agree that the scenarios you mentioned regarding **[Q3, Q4]** are excellent points, as they represent conditions where our proposed method is expected to particularly shine.
>
> - We would like to gently clarify the context of our current experiments. The settings we chose (e.g., $K\in \\{6,9,12\\}$ with $K'=3$) are already more demanding than those in our direct baseline, FFA-LoRA [1], which assumes all clients participate in every round ($K=K^\prime=3$).
>
>
> - Furthermore, while we agree that demonstrating scalability with a larger number of clients would be ideal, we encountered a **practical limitation** with the current dataset partitions. For instance, in our experiment with
> $K=12$ (`Fig. 4b`), some clients are allocated fewer training samples than the total number of data processed in one round (i.e., batch size × local steps $\tau$).
>
> - Scaling to a significantly larger client pool would magnify this issue and would likely require more extensive datasets to ensure each client has sufficient data for meaningful local training.
>
> - We believe that rigorously evaluating federated learning in these large-client regimes is a crucial and exciting future direction. Following your valuable suggestion, we will clarify this context in the revision and highlight it as a promising avenue for further work.
>
> - Thank you again for your insightful feedback.
>
> ---
>
> ### Reference
>
> [1] Sun, Youbang, et al. "Improving lora in privacy-preserving federated learning." ICLR 2024.

---

### Note · Authors · 2025-08-12

Dear reviewers and AC

We take this final remark as a valuable opportunity to express our sincere gratitude for your time and effort in overseeing and reviewing our paper. We would also like to outline how, upon acceptance, we plan to incorporate the concerns addressed during the rebuttal and discussion period into the revision:

---
>`Sec. 2`: Method
- **Motivation for SVD (yjb1)**: We will add several paragraphs after `L158` that discuss the motivation for recomputing SVD compared to FedRand, supported by experiments on FedRand.
- **Empirical supports on theoretical analysis (yjb1)**: We will include experiments conducted during the discussion period, which empirically support our theoretical analysis after `L191`.
---
>`Sec. 3`: Experiments
- **Challenging experimental setups (z79n, yjb1, and w3wU)**: We will emphasize our challenging experimental setups (i.e., stronger heterogeneity and larger client pools) compared to existing FL literature in `L220-229`, and add a pointer to `Appendix C` for detailed dataset statistics.
- **Complex Tasks (z79n and toPF)**: We will add experiments on SNLI and HellaSwag in `Sec. 3.2`.
- **Approximation of SVD (toPF, yjb1, and w3wU)**: We will add experiments on $\texttt{FedSVD}$ with an efficient approximation of SVD in `Sec. 3.3`.
- **DoRA (w3wU)**: We will add experiments, extending FedSVD to DoRA in `Sec. 3.3`.
- **DP budget (z79n and toPF)**: We will add experiments under a stricter privacy constraint ($\epsilon=3, \delta=10^{-5}$) in `Sec. 3.2`.
- **Upper bound (z79n)**: We will report the performance of a centrally, fully fine-tuned model as an upper bound in `Table 1, 2`.
- **Ablation on other datasets (yjb1)**: We will expand our ablation in `Table 3` to include other datasets, demonstrating the importance of orthnormality.
---
>`Sec. 4`: Conclusion
- **Optimality (w3wU)**: We will discuss the optimality of $\texttt{FedSVD}$ in the limitations section (`L320-329`).
---
>`Appendix`
- **FedPower, GaLore, and Gradient Subspace (toPF and yjb1)**: We will clarify how our approach differs from FedPower, GaLore, and Gradient Subspace following `L541` in `Appendix A`.
- **FFA-LoRA (w3wU)**: We will emphasize the differences between our method and FFA-LoRA in `L525` of `Appendix A`.
- **Dataset (z79n, yjb1, and w3wU)**: We will add a table showing specific dataset statistics per label for different clients in `Appendix C`.
---

Best regards,

The Authors

---

### Decision · Program_Chairs · 2025-09-17

**Decision:**

Accept (poster)

**Comment:**

This paper studies the private federated learning with LoRA. More specifically, the authors propose to reparameterize the LoRA update through SVD, which can effectively reduce the amount of the additive noise.

All the reviewers agree the proposed method is novel, and their concerns are successfully addressed during the discussion.

I encourage the authors to include the promised change and the new experiments (e.g., complex task, approximation of SVD, small privacy budgets, role of orthonormal) in the revision.